# Hydroxyl radicals dominate reoxidation of oxide-derived Cu in electrochemical $CO_2$ reduction

Shijia Mu[1], Honglei Lu[1], Qianbao Wu [1], Lei Li[1], Ruijuan Zhao[1], Chang Long [1✉] & Chunhua Cui [1✉]

$Cu^{\delta+}$ sites on the surface of oxide-derived copper (OD-Cu) are of vital importance in electrochemical $CO_2$ reduction reaction ($CO_2$RR). However, the underlying reason for the dynamically existing $Cu^{\delta+}$ species, although thermodynamically unstable under reductive $CO_2$RR conditions, remains uncovered. Here, by using electron paramagnetic resonance, we identify the highly oxidative hydroxyl radicals ($OH^\bullet$) formed at room temperature in $HCO_3^-$ solutions. In combination with in situ Raman spectroscopy, secondary ion mass spectrometry, and isotope-labelling, we demonstrate a dynamic reduction/reoxidation behavior at the surface of OD-Cu and reveal that the fast oxygen exchange between $HCO_3^-$ and $H_2O$ provides oxygen sources for the formation of $OH^\bullet$ radicals. In addition, their continuous generations can cause spontaneous oxidation of Cu electrodes and produce surface $CuO_x$ species. Significantly, this work suggests that there is a "seesaw-effect" between the cathodic reduction and the $OH^\bullet$-induced reoxidation, determining the chemical state and content of $Cu^{\delta+}$ species in $CO_2$RR. This insight is supposed to thrust an understanding of the crucial role of electrolytes in $CO_2$RR.

[1] Molecular Electrochemistry Laboratory, Institute of Fundamental and Frontier Sciences, University of Electronic Science and Technology of China, Chengdu 610054, China. ✉email: longch@uestc.edu.cn; chunhua.cui@uestc.edu.cn

Conversion of $CO_2$ into value-added chemicals through renewable electricity-powered electrochemical $CO_2$ reduction reaction ($CO_2RR$) has been recognized as a promising strategy to achieve "carbon-neutral"[1–3]. Oxide-derived copper (OD-Cu) has been proven as a group of efficient electrocatalysts for $CO_2RR$, especially for multi-carbon products ($C_{2+}$)[4–6]. The precise mechanism remains unknown and different views have been proposed[7–14]. Specifically, both experiments and theoretical calculations demonstrated that the $Cu^{\delta+}/Cu^0$ interface can activate the inert $CO_2$ molecules and promote CO-CO coupling[10,11]. To regulate the selectivity of $C_{2+}$ products, many efficient OD-Cu catalysts with characteristic $Cu^{\delta+}$ sites have been achieved through constructing various oxidized pre-catalysts or employing $CO_2$-pulsed electrolysis[7–9,12].

Thermodynamically, $CuO_x$ phases should be removed under the $CO_2RR$ conditions thereby the loss of the active $Cu^{\delta+}$ species[15]. While some studies have demonstrated the reduction of $CuO_x$ phases to metallic Cu during $CO_2RR$[16–20]. Interestingly, despite these, the $Cu^{\delta+}$ species has been frequently detected in $CO_2RR$[8,9,21–23]. This puzzling phenomenon leads to divergent views of the presence of $Cu^{\delta+}$ species. Cuenya et al. demonstrated that the $O_2$ plasma-treated $CuO_x$ is resistant to reduction[24]. Yu et al. found that in situ generated $CO_2RR$ intermediates on the surface of OD-Cu stabilize $Cu^{\delta+}$ species[8]. Chen et al. pointed out that the chemical states of Cu are associated with the oxidation caused by an uncertain oxidative species in the electrolytes[25–27]. We took the view that the $Cu^{\delta+}$ species should be dynamically existing, and we propose that the redox conditions provided by electrochemical cathodic reduction and oxidative species in electrolytes should be crucial. Understanding the reduction/reoxidation behavior of OD-Cu in commonly used $CO_2$-saturated $KHCO_3$ electrolytes is a grand challenge but vital to identifying what is the oxidative species.

Here, using in situ Raman spectroscopy, we observe the rapid reoxidation phenomenon of Cu to $Cu_2O$ species within a very short time scale (10 s) upon stopping the cathodic potential, and we record the dynamically existing $Cu^{\delta+}$ species at the surface of OD-Cu during the $CO_2RR$. We further identify that the rapid reoxidation is caused by strongly oxidative $OH^\bullet$ radicals existing in $KHCO_3$ solutions, by using electron paramagnetic resonance (EPR). With the isotope-labeling technique, we point out that the $OH^\bullet$ radicals are generated from both $HCO_3^-$ and $H_2O$ upon

oxygen exchange in $HCO_3^-$ aqueous solutions at room temperature. In addition, owing to the continuous generation of $OH^\bullet$ radicals, we observe higher degrees of oxidizing corrosion of Cu electrodes in $CO_2$- or Ar-saturated $KHCO_3$ solution under open circuit potential (OCP) relative to those electrolytes without $OH^\bullet$ radicals, giving a hint of oxidative $KHCO_3$ electrolytes. This work demonstrates unexpected $OH^\bullet$ radicals as the oxidative species, and it guides the fundamental understanding of the origin of $Cu^{\delta+}$ species in $CO_2RR$.

## Results

**Dynamic reduction/reoxidation behavior.** To enhance in situ Raman signals, a surface roughened OD-Cu electrode was selected as a model catalyst for this study[4,28]. It was prepared via depositing the micro-nano Cu particles onto the surface of the Cu mesh substrate (Supplementary Fig. 1), by using a modified electrodeposition method[29]. The as-prepared Cu electrodes show a mainly metallic Cu state, with surfaces being oxidized to $Cu_2O$ phases owing to the exposure to air after electrodeposition and $KHCO_3$ electrolytes before applying potentials for $CO_2RR$ (Supplementary Fig. 1).

We implemented potential-dependent Raman spectra to investigate the stability of surface $Cu_2O$ species under cathodic potentials in $CO_2$-saturated 0.5 M $KHCO_3$ solution (Supplementary Fig. 2). The vibration bands of surface $Cu_2O$ were observed at ~216, ~520, and ~620 cm$^{-1}$ at >0.0 V versus reversible hydrogen electrode ($V_{RHE}$)[28,30,31]. While, at <−0.2 $V_{RHE}$, the typical $Cu_2O$ signals disappear, indicating the surface $Cu^+$ species was reduced to metallic Cu. This result is supported by the prediction of the Pourbaix diagram for Cu and the previous reports[15–20]. In addition, the peaks at 282, 360, 2070–2100 cm$^{-1}$ are related to the frustrated $\rho$(Cu–C–O) rotational mode, $\nu$(Cu–CO) stretching mode, and intramolecular C≡O stretching vibration of CO intermediates, respectively. The bands at 2820–2950 cm$^{-1}$ are assigned to the -CH$_x$ stretching regions[30,31] from 0.2 to −0.6 $V_{RHE}$ (Fig. 1a, b and Supplementary Figs. 2–4), demonstrating the initiation of $CO_2RR$. Thus, to reduce surface $Cu_2O$ to metallic Cu (Supplementary Figs. 2 and 5) and to avoid the reconstruction of Cu at very negative potentials during $CO_2RR$[18,32], a moderate reduction potential of −0.3 $V_{RHE}$ was chosen for further Raman study.

The reoxidation of OD-Cu surface was investigated via real-time Raman test, by applying a reduction potential at −0.3 $V_{RHE}$ for 30 s firstly, and then switching to OCP for 60 s. The spectra were acquired every 10 s. As shown in Fig. 1a, once switching the potential from −0.3 $V_{RHE}$ to OCP, the metallic Cu was rapidly oxidized to $Cu_2O$ species within 10 s, and its three characteristic bands at 146, ~520, and ~620 cm$^{-1}$ re-appeared after 20 s[28,30,31]. This rapid reoxidation phenomenon indicates a strongly oxidative species existing in the $KHCO_3$ electrolyte, in contrast to the non-reoxidation process of OD-Cu in the Ar-saturated 0.25 M $K_2SO_4$ electrolyte (Supplementary Fig. 6). Here $K_2SO_4$ was selected as a control electrolyte because of its moderate solubility relative to $KClO_4$ (~0.12 M), suitable chemical stability, and relatively weaker interaction with Cu in contrast to such as KCl and KI.

To further confirm the reduction/reoxidation phenomenon, we implemented a loop test where alternate potentials between OCP for 20 s and −0.3 $V_{RHE}$ for 10 s were employed. As shown in Fig. 1b, during the five cycles, we observed that the $Cu_2O$ phase disappears at −0.3 $V_{RHE}$ and re-produces at OCP. We found that the surface Cu species go through the process: $Cu_2O \rightarrow CuO_x \rightarrow$ metallic Cu, with the cathode potential decreasing from OCP to −0.3 $V_{RHE}$ (Supplementary Fig. 2). It is a reverse process when switching from −0.3 $V_{RHE}$ to OCP. Thus, we suggest that the

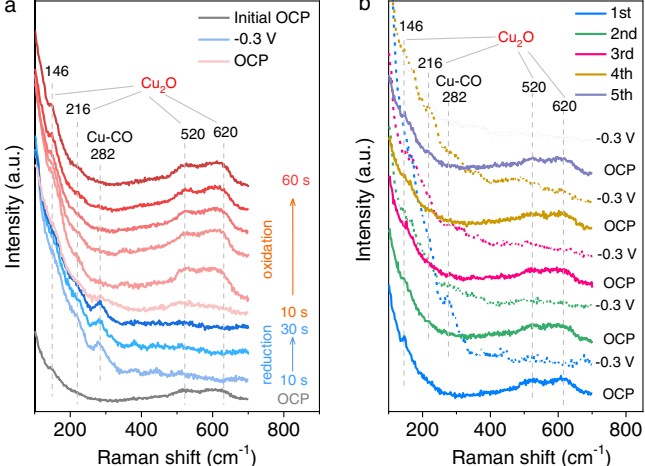

**Fig. 1 In situ Raman spectra of OD-Cu electrodes in $CO_2$-saturated 0.5 M $KHCO_3$. a** Real-time Raman spectra of surface $Cu_2O$ species at −0.3 $V_{RHE}$ and subsequently at open circuit potential (OCP). **b** Raman spectra under loop tests with a reduction potential at −0.3 $V_{RHE}$ for 10 s and reoxidation potential at OCP for 20 s.

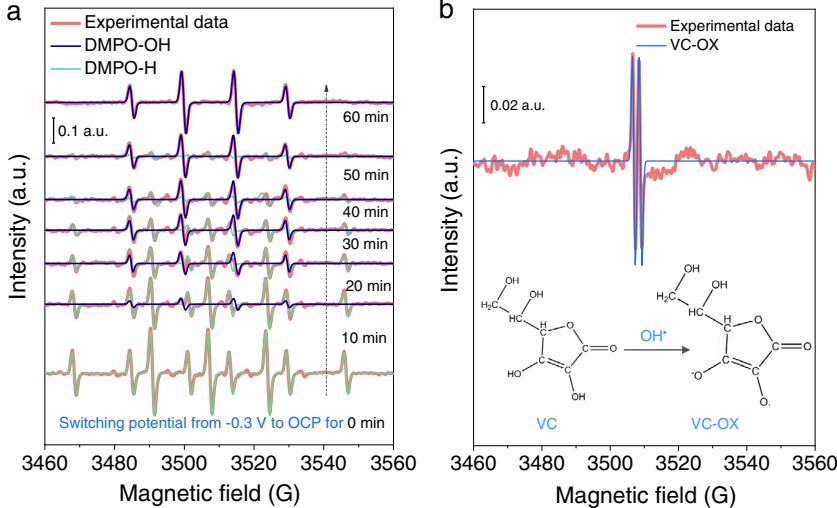

**Fig. 2 EPR spectra were recorded in a CO$_2$-saturated 0.5 M KHCO$_3$ electrolyte containing 100 mM DMPO. a** Time-dependent EPR spectra of the KHCO$_3$ solution after switching potential from $-0.3$ V$_{RHE}$ (20 min reduction) to OCP for different periods. **b** EPR spectra of the KHCO$_3$ solution with 10 mM VC. Inset shows the structure transition from VC and oxidized VC-OX caused by OH$^\bullet$. The simulated EPR spectra of DMPO-OH, DMPO-H, and VC-OX adducts were obtained according to their hyperfine splitting constants.

chemical state of Cu and the phase of surface Cu species are the results of dynamic equilibrium between the cathodic reduction and the reoxidation caused by strongly oxidative species in KHCO$_3$ electrolytes. This explanation may reconcile the debates on the stability of Cu$^{\delta+}$ species during CO$_2$RR observed from different groups[8,9,16–23].

To preclude the effect of trace undesired oxidative species, such as residual O$_2$, in the electrolyte or at the electrode surface, we implemented a long-time reduction of electrodeposited Cu electrode in CO$_2$-saturated KHCO$_3$ solution at $-0.3$ V$_{RHE}$ for 20 min. Then we switched the potential to OCP while recording the time-dependent optical microscopic images of the electrode surface (Supplementary Fig. 7). Within 10 min, the bright Cu surface gradually converted into a light-black surface, indicating an explicit oxidation process of Cu in the bulk KHCO$_3$ electrolyte.

**OH$^\bullet$ radicals in KHCO$_3$ electrolytes.** Inspired by the formation of various radicals in the KHCO$_3$ system[33–37], we speculate that there may be free radicals in the as-used KHCO$_3$ electrolytes. Thus, EPR spectroscopy was applied to investigate the radical species in a 0.5 M KHCO$_3$ aqueous solution. Considering the very short lifetimes of radicals (up to several ms), 5,5-dimethyl-1-pyrroline N-oxide (DMPO) as a spin trapping agent was added into KHCO$_3$ electrolytes, thus the formed DMPO-radical adducts have the lifetimes as long as minute-scale[38], facilitating the EPR tests.

To eliminate any pre-introduced oxidative species, a 20 min of long-time reduction on the Cu electrode at $-0.3$ V$_{RHE}$ was performed in CO$_2$-saturated 0.5 M KHCO$_3$ containing 100 mM DMPO. At $-0.3$ V$_{RHE}$, the hydrogen evolution reaction (HER) occurs and the generated hydrogen radicals (H$^\bullet$) can be trapped as a DMPO-H adduct (hyperfine splitting constants, A$_N$ = 1.65 mT, A$_H$ = 2.25 mT)[38]. Once the potential was switched to OCP, immediately the radicals in the solution were tested through a real-time EPR. As shown in Fig. 2a, DMPO-H radical adduct generated during the HER did not disappear as soon as we stopped the bias, owing to the increased lifetime, thus they can be measured even if we switched the potential from $-0.3$ V$_{RHE}$ to OCP. Interestingly, after 10 min, the DMPO-OH (A$_N$ = 1.50 mT,

A$_H$ = 1.48 mT)[39] emerged and gradually became the dominant signal with increasing the time within 60 min. Based on this carefully designed test protocol, we claim that the newly generated OH$^\bullet$ radicals were indeed activated in the HCO$_3^-$ electrolyte. The continuously ascending OH$^\bullet$ radical intensity was further confirmed when resting the KHCO$_3$ solution at OCP for 24 h without pre-reduction at $-0.3$ V$_{RHE}$ (Supplementary Fig. 8).

We assume that the generation of fresh H$^\bullet$ radical will end once stopping the potential at $-0.3$ V$_{RHE}$, thus it is reasonable that the DMPO-H signal decays owing to its limited lifetime (Fig. 2a). In contrast, the signal of DMPO-OH increased over 24 h (Supplementary Fig. 8), suggesting a continuous production of OH$^\bullet$ radicals with a considerable amount excited by the electrolyte, considering its half-time of DMPO-OH adduct (minute-scale)[40,41]. Besides, the Raman vibration of Cu-OH mode at 710 cm$^{-1}$, which could be caused by OH$^\bullet$ radical, was observed during CO$_2$RR from 0.2 to $-0.3$ V$_{RHE}$ (Supplementary Fig. 2)[28,42]. OH$^\bullet$ radical is a strongly oxidative species, with a high electrode potential of 2.73 V versus normal hydrogen electrode[43], which can lead to the rapid reoxidation of surface OD-Cu in KHCO$_3$ electrolytes during CO$_2$RR.

The control experiments showed that no relevant DMPO-OH signals were detected in both pure water and 0.25 M K$_2$SO$_4$ solutions containing 100 mM DMPO (Supplementary Fig. 9). Meanwhile, the reoxidation behavior of surface OD-Cu in 0.25 M K$_2$SO$_4$ electrolytes was not observed via the in situ Raman test (Supplementary Fig. 6). These results imply that pure water or K$^+$ cations alone cannot produce OH$^\bullet$ radicals at room temperature. Nevertheless, we found that HCO$_3^-$ anions play a key role in the generation of OH$^\bullet$ radicals, by tuning the HCO$_3^-$/SO$_4^{2-}$ mole ratio under the same K$^+$ concentration (0.5 M), where the DMPO-OH signals decreased with decreasing the HCO$_3^-$ concentrations (Supplementary Fig. 10).

To further verify the OH$^\bullet$ radicals, we added 10 mM vitamin C (VC) as an OH$^\bullet$ scavenger into the CO$_2$-saturated 0.5 M KHCO$_3$ solution containing 100 mM DMPO. As shown in Fig. 2b, the EPR signal of the DMPO-OH adduct disappeared and was replaced by a newly formed VC-OX radical (A$_H$ = 0.18 mT) that is from VC oxidation (VC-OX) by OH$^\bullet$ radicals[44]. The inset in Fig. 2b displays the oxidation process. The oxidation phenomenon was further demonstrated via the color variation of VC

KHCO₃ solution from colorless to light yellow after 24 h aging (Supplementary Fig. 11). A control experiment excludes the VC-OX radicals from self-oxidation, where no EPR signals were observed in the 10 mM VC water solution containing 100 mM

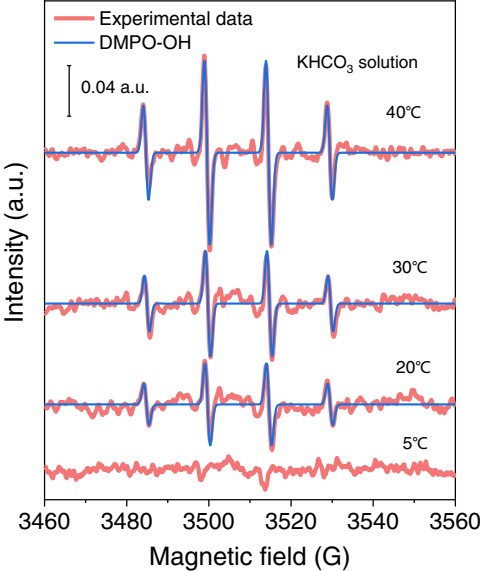

**Fig. 3 Temperature-dependent EPR spectra from 5 to 40 °C.** EPR data were recorded in Ar-saturated 0.5 M KHCO₃ solutions containing 100 mM DMPO without pre-reduction operations. Each spectrum was acquired after 8 h resting at the indicated temperature.

DMPO (Supplementary Fig. 12). Besides, the reoxidation of OD-Cu electrode at OCP was not observed after the reduction of surface Cu₂O species to metallic Cu at −0.3 V_RHE in the CO₂-saturated KHCO₃ solution containing VC, by using in situ Raman spectroscopy (Supplementary Fig. 13).

To consider whether the formation of OH• radicals could be a thermally activated process, the signal of the DMPO-OH adduct has been tracked via a temperature-dependent study. We hypothesize that the room temperature may activate HCO₃⁻ solutions to produce OH• radicals. To confirm it, we implemented the temperature-dependent EPR measurements from 5 to 40 °C in an Ar-saturated 0.5 M KHCO₃ electrolyte containing 100 mM DMPO (Fig. 3). When the temperature is as low as 5 °C, no EPR signals were detected. An obvious EPR signal from DMPO-OH was observed around 20 °C and increased with the enhanced temperature. This result indicates that temperature is a key parameter for OH• radical generation in the KHCO₃ solutions.

**Oxidizing corrosion of Cu plate in KHCO₃ solution.** Given the steady generation of OH• radicals in KHCO₃ electrolytes, we supposed that it would result in a higher degree of oxidizing corrosion of Cu metal. A polished Cu plate (inset in Fig. 4a) was placed into a CO₂-saturated KHCO₃ solution. After 24 h resting, a visually light-black Cu surface was observed (inset in Fig. 4b). To preclude the CO₂(aq) effect, the same operation was implemented in Ar-saturated KHCO₃, and a darker Cu surface was observed associated with a stronger DMPO-OH signal (inset in Fig. 4c). Thus, the Cu oxidation by CO₂(aq) was ruled out. To further ascertain Cu oxidation by electrolyte-induced OH•, the

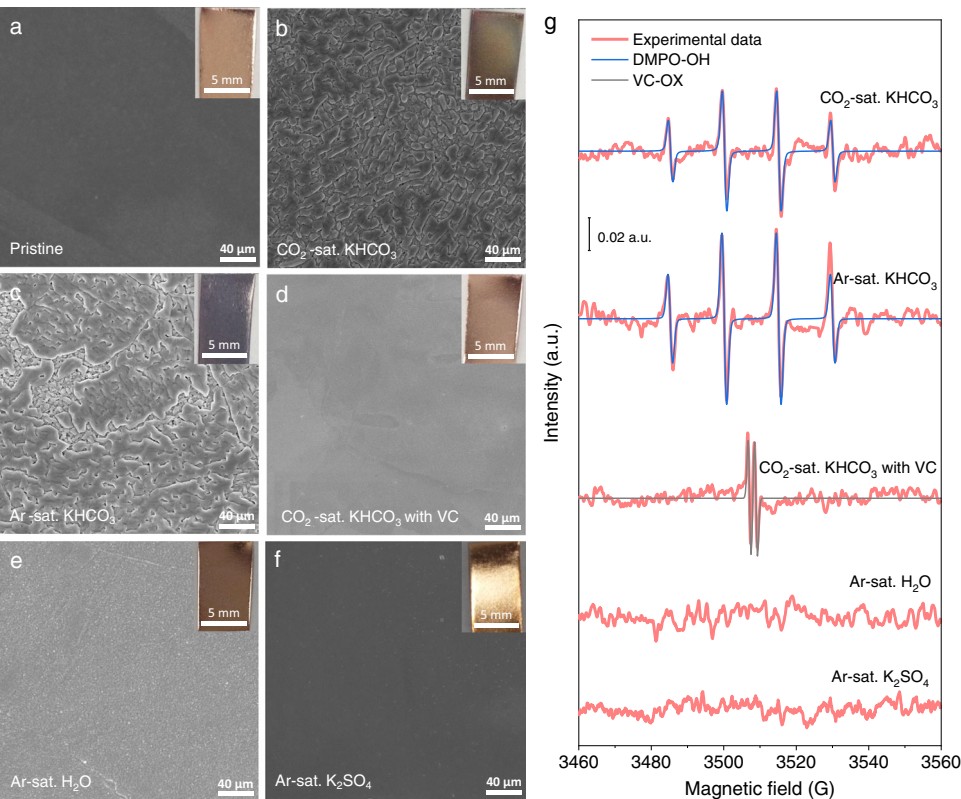

**Fig. 4 Oxidizing corrosion of the polished Cu plates in different solutions.** SEM images of Cu plates before **a**, and after 24 h oxidizing corrosions in **b** CO₂-saturated KHCO₃, **c** Ar-saturated KHCO₃, **d** CO₂-saturated KHCO₃ containing VC, **e** Ar-saturated ultrapure water, and **f** Ar-saturated K₂SO₄ solutions. Correspondingly, the photographs are displayed in insets. **g** EPR spectra of the corresponding solutions containing 100 mM DMPO after 24 h resting. The concentrations of KHCO₃, K₂SO₄, and VC are 0.5 M, 0.25 M, and 10 mM, respectively.

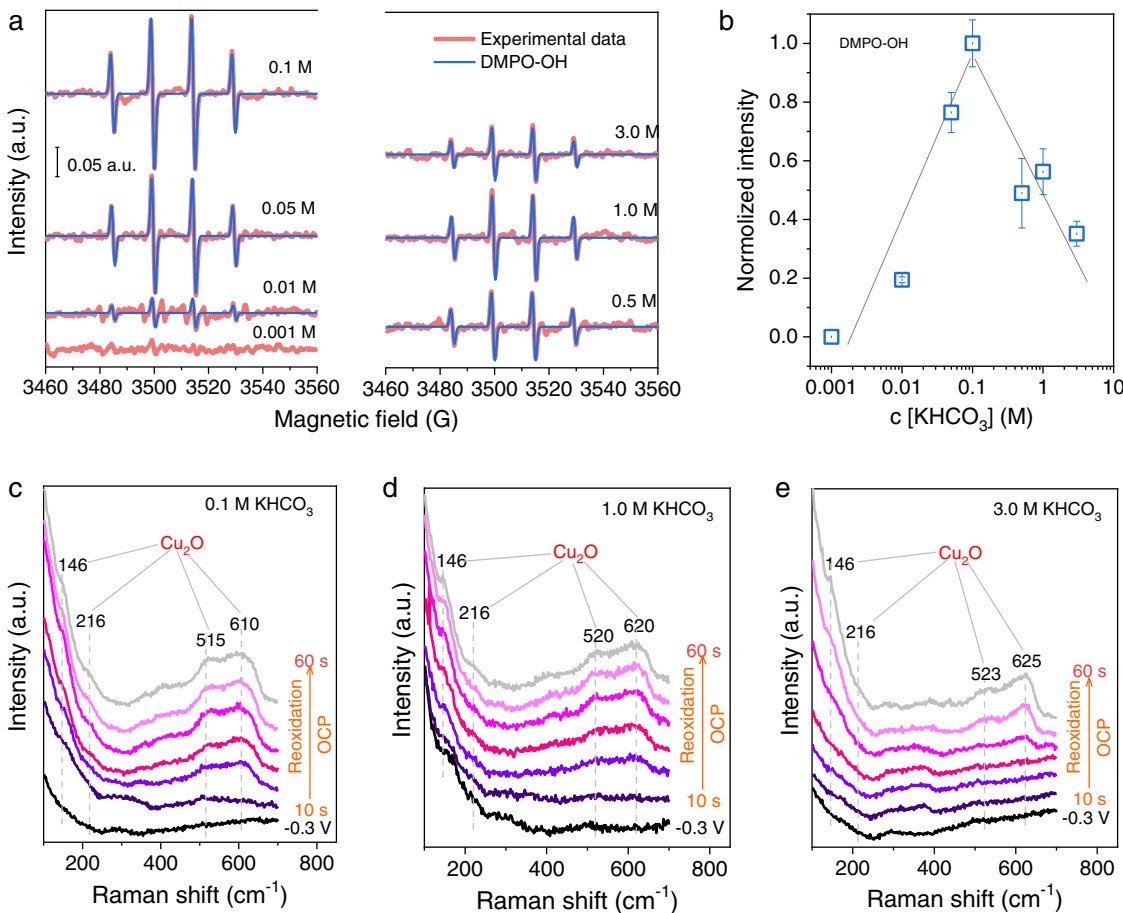

**Fig. 5 Relationships between HCO₃⁻ concentration, OH• radical amount, and reoxidation dynamics of OD-Cu electrodes. a** The acquired EPR spectra in different $HCO_3^-$ concentrations. Each solution contains 100 mM DMPO. **b** Correspondingly, the normalized intensity of DMPO-OH versus $HCO_3^-$ concentration. The intensity was normalized by that of 0.1 M $HCO_3^-$. The gray lines guide the trends of DMPO-OH intensity without the compensation of $K^+$. The error bars represent the standard deviation. **c–e** Raman spectra of OD-Cu reoxidation in different KHCO₃ concentrations at OCP after reduction at $-0.3$ V$_{RHE}$.

control experiments were carried out in KHCO₃ solution containing VC, pure water, and K₂SO₄ solution, respectively. No obvious color changes were visually seen for the three cases (insets in Fig. 4d–f), compared to the pristine Cu plate (inset in Fig. 4a).

The surface morphology of polished Cu plates before and after oxidizing corrosions was characterized by using an optical microscope (Supplementary Fig. 14) and scanning electron microscopy (SEM, Fig. 4a–f). The oxidized surfaces were observed in CO₂- and Ar-saturated KHCO₃ solutions (Fig. 4b, c), relative to pristine Cu (Fig. 4a), KHCO₃ solution with VC (Fig. 4d), pure water (Fig. 4e), and K₂SO₄ solution (Fig. 4f). Moreover, SEM images showed that the thicknesses of oxide layers at Cu plates reach ~596 and ~780 nm in CO₂- and Ar-saturated KHCO₃ solutions, respectively (Supplementary Fig. 15). The phase analyses by X-ray diffraction (Supplementary Fig. 16) and high-resolution transmission electron microscopy (HRTEM, Supplementary Fig. 17) illustrated that the corrosion products mainly consist of Cu₂O. After corrosions, the atomic content of the O element determined by energy-dispersive spectroscopy (Supplementary Figs. 18–23) increased from pristine 0.65% to 14.72% and 19.82% in CO₂- and Ar-saturated KHCO₃ solutions respectively, while no obvious variations were discerned for the other three cases (Supplementary Fig. 24).

The oxidizing corrosion degrees are well related to the amount of OH• radicals. As shown in Fig. 4g, after 24 h resting, the CO₂-

and Ar-saturated KHCO₃ solutions display obvious EPR signals of DMPO-OH and the latter has a higher intensity. By comparison, there was no signal for the other three solutions, except for the KHCO₃ solution with VC that presents an EPR signal of VC-OX due to VC oxidation by OH• radicals. These results are further supported by analyzing the surface chemical states of Cu 2$p$ of oxidized Cu plates using X-ray photoelectron spectroscopy (XPS), where a little Cu²⁺ species with deep oxidation was found for CO₂- and Ar-saturated KHCO₃ solutions, while only Cu¹⁺/Cu⁰ species were identified for the other cases (Supplementary Fig. 25).

**Sources of OH• radicals in KHCO₃ solutions.** According to the above analysis, $HCO_3^-$ anions play a vital role in determining the OH• generation. We thus studied the relationships between $HCO_3^-$ concentration, the intensity of OH• radicals, and reoxidation dynamics on OD-Cu electrodes. Firstly, we investigated the influence of $HCO_3^-$ concentrations on the amount of OH• radicals. As shown in Fig. 5a, at <0.1 M, the DMPO-OH signal increases with increasing the $HCO_3^-$ concentrations, yet further increasing the concentrations cannot produce more OH• radicals. It is worth noting that further increasing the $HCO_3^-$ concentration enhances the $K^+$ concentration as well. Thus, at the same $K^+$ concentration (0.5 M) with $K^+$ compensation by K₂SO₄, the relationship between the $HCO_3^-$ concentration and the intensity of DMPO-OH is more pronounced following the order

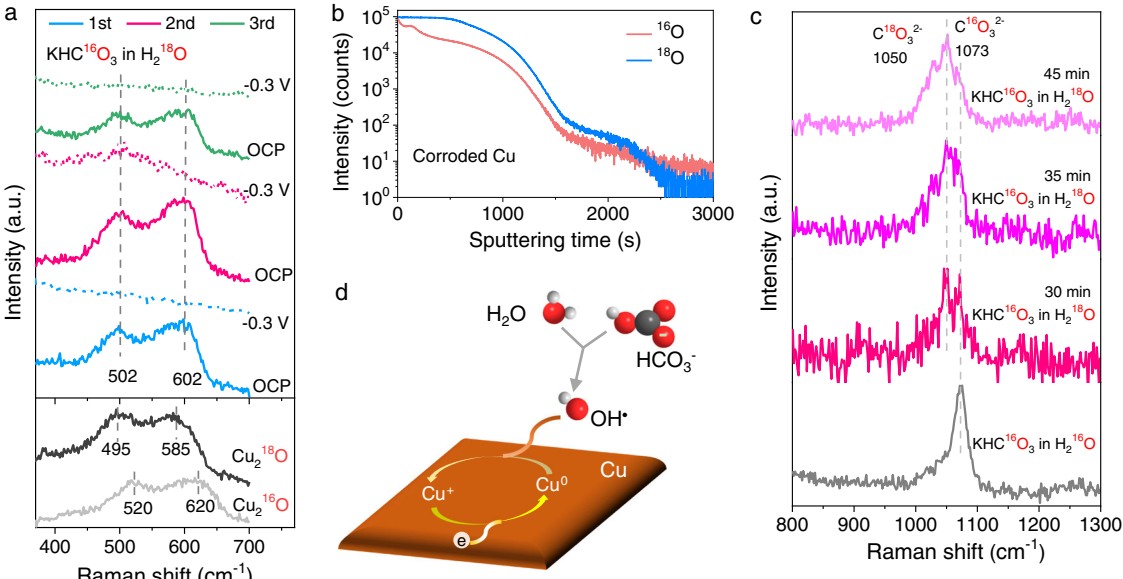

**Fig. 6 Determination of OH• sources by oxygen isotope labeling in Ar-saturated 0.5 M KHC$^{16}$O$_3$ H$_2$$^{18}$O solution. a** Raman spectra of surface Cu$_2$O species under the loop tests with reduction at −0.3 V$_{RHE}$ for 10 s and subsequently reoxidation at OCP for 20 s. The plots at the bottom show reference spectra for Cu$_2$$^{16}$O and Cu$_2$$^{18}$O. **b** $^{16}$O and $^{18}$O contents of the CuO$_x$ species at the Cu plates after 24 h oxidizing corrosion, measured by TOF-SIMS. **c** Raman spectra of carbonate species on the Cu electrode. Each spectrum was obtained after resting the solution for the indicated time. The bottom spectrum shows C$^{16}$O$_3$$^{2−}$ vibration as a reference. **d** The proposed dynamic reduction/reoxidation process of surface OD-Cu during CO$_2$RR.

0.2 M < 0.4 M ≤ 0.5 M (Supplementary Fig. 10), and the intensity of DMPO-OH tends to saturate. The optimal HCO$_3$$^−$ concentration without K$^+$ compensation for the OH• radical formation is around 0.1 M (Fig. 5b).

Then, to correlate the OH• radicals with reoxidation of OD-Cu electrodes, we carried out in situ Raman in different concentrations of HCO$_3$$^−$ solutions. As shown in Fig. 5c–e, with increasing the HCO$_3$$^−$ concentration from 0.1 to 3.0 M, the Raman intensity of Cu$_2$O species decreases, and the reappearance of Cu$_2$O phase delays, during the reoxidation processes after reduction at −0.3 V$_{RHE}$. Thus, it is indeed that the intensity of DMPO-OH is very related to the reoxidation ability. These results suggest that HCO$_3$$^−$ contributes to the generation of OH• radicals that oxidize the Cu surfaces.

To clarify the sources of OH• radicals, we applied a Raman measurement with oxygen isotope labeling in Ar-saturated 0.5 M KHC$^{16}$O$_3$ H$_2$$^{18}$O solution. The Raman vibrations of surface Cu$_2$O species were recorded again by applying alternate potentials between −0.3 V$_{RHE}$ and OCP. As shown in Fig. 6a, during the three cycles, the characteristic bands of Cu$_2$O species at ∼502 and ∼602 cm$^{−1}$ locate between those of Cu$_2$$^{16}$O (520, 620 cm$^{−1}$) and Cu$_2$$^{18}$O (495, 585 cm$^{−1}$). This indicates that both HC$^{16}$O$_3$$^−$ anions and H$_2$$^{18}$O are responsible for the OH• generation.

To further reveal the respective contributions of HC$^{16}$O$_3$$^−$ and H$_2$$^{18}$O to OH• radicals, we implemented time-of-flight secondary-ion mass spectrometry (TOF-SIMS) measurement to investigate the $^{16}$O and $^{18}$O contents of CuO$_x$ species formed after 24 h oxidizing corrosion of Cu plate in Ar-saturated 0.5 M KHC$^{16}$O$_3$ H$_2$$^{18}$O solution. As illustrated in Fig. 6b, both $^{16}$O and $^{18}$O are detected in the whole sampling depth, and the latter has a little higher content. Concerning that pure water does not donate OH• radicals (Fig. 4g and Supplementary Fig. 9), and the isotope oxygen mole ratio (HC$^{16}$O$_3$$^−$ over H$_2$$^{18}$O is around 1:100), we hypothesize that oxygen exchange between H$_2$$^{18}$O and HC$^{16}$O$_3$$^−$ may exist.

To support this hypothesis, we tracked the adsorbed carbonate species on the OD-Cu electrode in Ar-saturated KHC$^{16}$O$_3$ H$_2$$^{18}$O solution by using Raman spectroscopy. As displayed in Fig. 6c,

within 30 min, both C$^{16}$O$_3$$^{2−}$ vibration from pristine HC$^{16}$O$_3$$^−$ at 1073 cm$^{−1}$ and C$^{18}$O$_3$$^{2−}$ vibration from $^{18}$O-derived HC$^{18}$O$_3$$^−$ at 1050 cm$^{−1}$ were detected[45], indicating a fast oxygen exchange. After 45 min, dominant C$^{18}$O$_3$$^{2−}$ vibration from $^{18}$O-exchanged HC$^{18}$O$_3$$^−$ was observed. Thus, we show that the OH• radicals have been generated at room temperature in HCO$_3$$^−$ electrolytes while the dynamic oxygen exchange between HCO$_3$$^−$ and H$_2$O supplies oxygen sources for the formation of OH• radicals.

Based on the above results, we proposed a mechanism for the dynamic reduction/reoxidation behavior at the OD-Cu surface during CO$_2$RR in Fig. 6d. The reduction of Cu$^{δ+}$ species to metallic Cu$^0$ driven via cathodic reduction competes with the reoxidation of Cu$^0$ to Cu$^{δ+}$ state caused by the highly oxidative OH• radicals. Thereby, there is a "seesaw-effect" between the reduction and reoxidation, determining the chemical state of Cu and the content of CuO$_x$ species at the surfaces of Cu electrodes in CO$_2$RR.

We discovered that strongly oxidative OH• radicals can be easily generated in HCO$_3$$^−$ aqueous solutions at room temperature, and the fast oxygen exchange between HCO$_3$$^−$ and H$_2$O provides dynamic oxygen sources for the OH• radical formation. The generated OH• radicals enable rapid reoxidation of OD-Cu electrodes in KHCO$_3$ electrolytes during CO$_2$RR. Besides, the continuous generation of OH• radicals can make higher degrees of oxidizing corrosion of Cu electrodes in KHCO$_3$ solutions to form substantial surface CuO$_x$ species relative to those electrolytes without OH• radicals. We further suggest that the dynamic chemical states of Cu and the content of surface CuO$_x$ species are determined by a "seesaw-effect" between the cathodic reduction potentials and the OH• radical-involving oxidation. This work provides insights into the reoxidation mechanism of OD-Cu and a general guide for understanding the crucial role of electrolyte composition for the CO$_2$RR.

## Methods

**Electrodeposited Cu electrodes**. All the electrochemical operations were performed using a Bio-logic SP200 potentiostat. A modified electrodeposition method was used to deposit micro-nano Cu particles at the Cu mesh substrate[29]. The copper mesh was ultrasonically cleaned in acetone, ethanol, and deionized water in

sequence. After drying by flowing $N_2$, the electrodeposition was carried out by applying a reduction current of $-40$ mA cm$^{-2}$ for 20 min to the Cu mesh electrode in an Ar-saturated solution consisting of 0.1 M CuSO$_4$·5H$_2$O (>99.99%, Aladdin) and 1.5 M H$_2$SO$_4$ (>98%, Chron Chemicals). The as-prepared electrode was rinsed with water and ethanol sequentially and then dried under a stream of $N_2$.

**Polished Cu plates**. Cu plate with a thickness of ~0.2 mm was first polished by using 3000 mesh sandpaper and then cleaned in acetone, ethanol, and deionized water in sequence. Further, it was electropolished in 85% phosphoric acid (Chron Chemicals) solution at 3.0 V versus counter-electrode of another Cu plate for 5 min. After that, the polished Cu plate was ultrasonically cleaned in Ar-saturated ultrapure water (Milli-Q, 18.2 MΩ) to remove the surface residual particles.

**Oxidizing corrosion of polished Cu plates**. The polished Cu plates were placed into CO$_2$-, Ar-saturated 0.5 M KHCO$_3$, CO$_2$-saturated 0.5 M KHCO$_3$ containing 10 mM VC, Ar-saturated ultrapure water, Ar-saturated 0.25 M K$_2$SO$_4$, and Ar-saturated 0.5 M KHCO$_3$ H$_2$$^{18}$O solutions, respectively. After resting for 24 h, the corrosive Cu plates were rinsed with water and then dried under a stream of $N_2$.

**Materials characterization**. Crystal phase structures were characterized by an XRD diffractometer (Equinox 1000, Thermo Fisher Scientific) with Cu Kα radiation ($\lambda = 1.54$ Å). Morphology was observed by field-emission SEM (FEI Inspect F50). Elemental analysis was implemented by using EDS. TEM and HRTEM images were acquired by using an FEI Titan G2 60-300 electron microscope. The surface chemistry of the Cu electrode/plate was investigated using XPS (Thermo ESCALAB 250XI) with Al Kα X-rays (1486.6 eV). The depth $^{16}$O and $^{18}$O contents of CuO$_x$ species formed at the surface of the Cu plate were analyzed by applying TOF-SIMS (ION TOF-SIMS 5) with 30 keV-Bi$_3^+$ as an analysis gun and 2 keV-Cs$^+$ as the sputtering source.

**In situ Raman spectroscopy**. Raman spectra were recorded with an XploRA PLUS Raman spectrometer (Horiba Jobin Yvon) equipped with a ×50 objective and a 638 nm He-Ne laser. The filter was set at 50%. The measurements were conducted using a custom-made three-electrode electrochemical cell with a quartz window, in which the as-prepared Cu electrode, Ag/AgCl (3.0 M KCl), and membrane-separated Pt wire were used as the working, reference, and counter electrodes. Before each test, the as-used electrolyte was pumped into the cell at a rate of 2 ml min$^{-1}$. The equipped optical microscope was applied to acquire the real-time microscopic images of as-used Cu electrodes during the Raman tests.

**EPR spectroscopy**. EPR measurements were carried out using a continuous-wave Bruker EMX micro spectrometer operating in X-band mode with a frequency of 9.848 GHz at room temperature. Each spectrum was recorded using the following parameters: a microwave power of 20 mW, modulation amplitude of 1.0 G, and a single scan with a sweep time of 5 min. DMPO (Dojindo) was selected as the spin trapping agent, and its concentration in all solutions was 100 mM. For EPR measurements taken during electrocatalysis in the presence of a Cu electrode, the electrolyte was immediately measured after electrolysis at a given potential in a specific electrolyte containing 100 mM DMPO. EPR simulations were performed according to the hyperfine splitting constants of radicals, via using the Xenon software provided by the EPR manufacturer.

## Data availability

All the data that support the findings of this study are available within the paper and its Supplementary Information files, or from the corresponding author on reasonable request.

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

## Acknowledgements

C.C. acknowledges funding support from the Natural Science Foundation of China (22072013). S.M. thanks funding support from the China Postdoctoral Science Foundation (No. 2020M673169).

## Author contributions

C.C. supervised the project. S.M. carried out the experiments. H.L. and Q.W. contributed to the EPR measurements and analysis. L.L. and R.Z. contributed to partial Raman measurements. S.M., C.L. and C.C. wrote and revised the manuscript. All authors commented on the manuscript.

## Competing interests

The authors declare no competing interests.
