## [Peer Review File · Nature Communications]

Reviewers' comments:

Reviewer #1 (Remarks to the Author):

This research suggests that the presence of OH[·] radical in the HCO₃⁻ solution is the key to causing CuO_x species during CO₂RR. However, the presence of OH[·] was detected by using ex situ electron paramagnetic resonance. The result can not support the claim that OH[·] radical will cause the oxidation of Cu during CO₂RR. Most importantly, the author did not detect any oxide species under cathodic potential by using in situ Raman spectroscopy (Figure 1). Moreover, Raman spectroscopy is widely known as surface-sensitive technique. The Raman results obviously contradict the authors' claim that the OH[·] radical affects the chemical state of Cu during CO₂RR. This research only can conclude that Cu is prone to oxidize under KHCO₃ electrolyte compared to other conditions under OCP due to the presence of OH[·] radical. Based on the aforementioned reasons, this research did not meet the requirement for publishing in Nat. Commun.. The followings are more suggestions:

1. In the manuscript, the authors try to eliminate the undesired oxidative species by applying cathodic potential. For instance, in line 101, "To preclude the effect of a trace amount of undesired oxidative species, such as residual O₂ in the electrolyte or at the electrode surface, we implemented a long-time reduction with Cu electrode in CO₂ saturated KHCO₃ solution at -0.3 VRHE for 5 min." and in line 112, "To eliminate the pre introduction of any oxidative groups, a 20 min of long time reduction on the Cu electrode at 0.3 V RHE was performed in CO₂ saturated 0.5 M KHCO₃ containing 100 mM DMPO.". The authors should unify the conditions for eliminating oxidative species. Most importantly, the authors should prove that they chose the right condition for eliminating any oxidative species in the electrolyte.

2. The authors carried out the control experiments in both pure water and K₂SO₄ to show that HCO₃⁻ is the key for generating OH radicals. However, I consider that it is much crucial to show that Cu did not oxidize in these conditions. For instance, the authors should carry out the in situ Raman experiments mentioned in Fig. 1 for the conditions regarding in line 130-136.

3. Regarding the content mentioned in line 164 to line 167, "To further ascertain Cu oxidation associated with electrolyte-induced OH[·], the control experiments were carried out in KHCO₃ solution containing VC, pure water, and K₂SO₄ solution, respectively. No obvious color changes were visually seen for the three cases (insets in Fig. 3 d f), compared to the pristine Cu plate (inset 167 in Fig. 3 a)" by visually observing the samples is not a reliable method to ascertain the claim. The authors should deploy Raman spectroscopy to investigate the samples since the Raman technique is surface sensitive.

4. According to the authors' Raman results, it is metallic Cu conducting the CO₂RR reaction. There were no evidence in this research showing that the oxide species can retain during the reaction. As a result, I do not consider that the authors can suggest the presence of Cu^{δ+} species during CO₂RR.

5. Since the authors try to understand whether the oxide species of oxide-derived copper can retain during CO₂RR, the author should carry out the experiments by using "oxide-derived copper catalyst" instead of Cu plate. Reference such as, J. Am. Chem. Soc. 2012, 134, 17, 7231–7234. would make the result more persuasive.

Reviewer #2 (Remarks to the Author):

S. Mu et al. examine experimentally the oxidation of Cu electrocatalyst in CO₂-saturated aqueous electrolytes. By using a trapping strategy, they provide evidence of the formation of OH radicals, to which they attribute the re-oxidation of Cu at the open circuit potential (OCP).

Overall, the work is an important and significant contribution to the study of Cu electrocatalyst employed for CO₂ reduction. As the authors accurately state in the Introduction, the prior literature has what might appear to be conflicting reports ranging from complete reduction of Cu oxides under CO₂R conditions vs. evidence that some oxides, presumably metastable, persist at relatively negative potentials. In this context, authors provide a plausible mechanism for the formation of such oxide layers. Moreover, they posit that there might be a “see-saw” effect in which there is an interplay between the rate of the reduction reactions and the formation of the oxidizing OH radicals.

Technically, the work is excellent, with impressive and thorough characterization. Raman spectroscopy clearly show the re-oxidation (Figure 1) and I found the use of isotopic labeling to be elegant in supporting the peak assignments (Figure 4). The scavenging experiments using vitamin C are convincing in implication OH radical in the re-oxidation (dark oxidation experiments are also convincing, inset of Figure 3b)

I support publication of this ms. subject to the following minor revisions.

Authors must discuss the thermodynamic plausibility of the carbonate/water “interplay” (line 230) that produces the OH radicals, even if they do not have a calculation to support their reasoning. Indeed, as the authors state in the Introduction, “CuO_x phases should be removed under the CO₂RR operations thereby [causing] the loss of the active Cu^{δ+} species.” Thus an observation that would seem to challenge this thermodynamic view requires some additional comment and, if possible, an explanation. A similar comment applies to the observation of H radicals in the bicarbonate electrolyte solution, page 5.

In the same vein, is the presence of Cu (or a metal surface) required to generate the radicals? That is, the experiments described in lines 108-129 have a Cu electrode presence. Would radicals be made in a similar amount if the metal were not present?

Minor revisions, not technical in nature.

In Figure 2 (and in most of the EPR spectra), the fits shown in color are more prominent than the data, which is shown with a thin line in light grey. The figure should be modified such that the experimental data is more prominent than the fit.

Line 179, replace “corrosives” with “corrosion products.”

Line 147, replace “OH adduct disappeared, instead of a” with OH adduct disappeared and was replaced by”

Reviewer #3 (Remarks to the Author):

The major claim of this manuscript is that during CO₂RR, HCO₃⁻ and H₂O reacts to give OH radicals. These OH radicals are proposed to oxidize the Cu surface during CO₂RR. Experimental evidence presented include the use of in situ Raman spectroscopy, secondary ion mass spectrometry, and isotope-labelling.

Overall, I do not agree with the claims of this study, as presented in the manuscript. There are major gaps that need to be addressed, before this manuscript can be re-considered.

Major:

- Authors claimed that HCO₃⁻ and H₂O reacts to give OH radicals. And the OH radicals are responsible for oxidizing the Cu surface. But this does not make sense, since in other electrolytes such

as KOH (without any HCO_3^- present), the Cu surfaces will also quickly oxidize once cathodic potentials are removed.

- Pg 5- Are the authors suggesting that HER occurs at OCP on Cu? How can this be? What is the OCP? Is H_2 gas detected? Can the authors use gas chromatography to verify this?

- What is the lifetime and stability of the OH radical? How can it be so long that its signal keeps increasing (Suppl Fig 5)?

- Could a HCO_3^- concentration dependent studies be done? I will assume that if the authors are correct, a higher HCO_3^- concentration will lead to faster oxidation of the Cu surface.

- What is the mechanism for OH radical formation from $\text{HCO}_3^- + \text{H}_2\text{O}$?

- End of pg 6 and start of Pg 7: The authors do not realize that water can also serve as a source of O to oxidize Cu. It does not need to be O_2 gas or OH radical (if these do exist).

Minor:

- In the introduction of the manuscript, the authors wrote statements such as 'This is due to the partially charged $\text{Cu}\delta^+$ species that plays a crucial role during CO_2RR ', ' $\text{Cu}\delta^+$ species has been frequently detected during CO_2RR .' etc. I think this is an unbalanced introduction. The existence of $\text{Cu}\delta^+$ species during CO_2RR is still controversial. Papers that claim that this species exist often based their claim on results obtained from ex-situ methods. Since Cu oxidizes so easily, the data obtained from ex-situ methods are questionable. I suggest that the authors try to balance out their discussion.

- Fig 1: I suggest that authors show their spectra to at least 2500 cm^{-1} , so that the C-O stretching vibration of $\text{CO}(\text{ad})$ at about 2000 cm^{-1} can be seen.

Dear Reviewers:

We appreciate your careful review and constructive comments which help us to significantly improve the quality of this manuscript (NCOMMS-21-46259). Now we have carefully considered all your comments and revised the manuscript accordingly. We made corresponding corrections point by point as follows (corrections in the revised manuscript are marked in blue color).

Reviewer #1 (Remarks to the Author):

This research suggests that the presence of OH^\cdot radical in the HCO_3^- solution is the key to causing CuO_x species during CO_2RR . However, the presence of OH^\cdot was detected by using ex situ electron paramagnetic resonance. The result cannot support the claim that OH^\cdot radical will cause the oxidation of Cu during CO_2RR . Most importantly, the author did not detect any oxide species under cathodic potential by using in situ Raman spectroscopy (Figure 1). Moreover, Raman spectroscopy is widely known as surface-sensitive technique. The Raman results obviously contradict the authors' claim that the OH^\cdot radical affects the chemical state of Cu during CO_2RR . This research only can conclude that Cu is prone to oxidize under KHCO_3 electrolyte compared to other conditions under OCP due to the presence of OH^\cdot radical. Based on the aforementioned reasons, this research did not meet the requirement for publishing in Nat. Commun. The followings are more suggestions:

Responses 1:

- We highly appreciate the suggestions to improve the quality of this paper. We are sorry that we did not clearly describe the key points leading to misleading.
- Regarding electron paramagnetic resonance (EPR) tests, the final goal of this work is to clarify whether the HCO_3^- electrolyte can activate OH^\cdot radicals via the excitation of room temperature. Here two points should be considered: (1), The excitation of OH^\cdot radicals does not need to apply cathodic potential yet the production of such as H^\cdot radicals require cathodic potentials. It means that the formation of OH^\cdot radicals in the bulk electrolyte can not be influenced by the cathodic potentials except for the situation at the electrode interface, where the electrolyte-activated OH^\cdot could either react with H^\cdot or get one electron from the electrode to form OH^\cdot . Thus, in principle, we just need to test electrolytes with EPR directly, but we cannot exclude the pre-introduced oxidative species, such as O_2 , or pre-existing OH^\cdot radicals. (2), So to confirm that the OH^\cdot radicals were newly generated, we applied cathodic potential at -0.3 V for 20 min. While we added DMPO as a spin trapping agent to increase the lifetime of radicals from ms to tens of mins. Then the applied cathodic potential and the electrochemically generated DMPO-H can gradually eliminate pre-introduced oxidative species, with an indicator of only the DMPO-H signal. Then we stopped the cathodic potential, and after that, we claim that the newly generated DMPO-OH signal was indeed activated in the HCO_3^- electrolytes. We added one sentence on pages 5 and 6 in the main text as follows:

“Based on this carefully designed test protocol, we claim that the newly generated OH^\cdot radicals were indeed activated in the HCO_3^- electrolyte.”

- We have demonstrated the strongly oxidizing OH^\cdot radicals in the HCO_3^- aqueous solutions that lead to the rapid reoxidation of OD-Cu electrodes. However, the argument is whether the reoxidation behavior takes place during CO_2RR . As suggested, we supplemented *in-situ Raman* to detect the oxidative species at the surface of the OD-Cu electrode (Figure R1). The Raman spectra show such as the newly formed Cu-OH vibrations at 710 cm^{-1} , which could be caused by OH^\cdot radical, during CO_2RR from 0.2 to $-0.3\text{ V}_{\text{RHE}}$. The

supplementary Raman is shown in Figure S2 and Figure S3 on page S3 and page S4 in the *Supporting Information* and correspondingly the discussion has been added on pages 3 and 6 in the main text as follows:

“In addition, the peaks at 282, 360, 2070-2100 cm^{-1} are related to the frustrated $\rho(\text{Cu}-\text{C}-\text{O})$ rotational mode, $\nu(\text{Cu}-\text{CO})$ stretching mode, and intramolecular $\text{C}\equiv\text{O}$ stretching vibration of CO intermediates respectively. The bands at 2820-2950 cm^{-1} are assigned to the $-\text{CH}_x$ stretching regions from 0.2 to -0.6 V_{RHE} (Fig. 1a and b, Supplementary Figs. 2-4)^{30,31}, demonstrating the occurrence of CO_2RR .”

“Besides, the Raman vibration related with Cu-OH mode at 710 cm^{-1} , which could be caused by OH⁻ radical, was observed during CO_2RR from 0.2 to -0.3 V_{RHE} (Supplementary Fig. 2)”

Figure R1. Raman spectra of surface species at OD-Cu electrode under electrolysis at indicated potentials versus RHE in a CO_2 -saturated 0.5 M KHCO_3 solution. (a) Vibration bands of the surface Cu_2O , generated intermediate CO, and surface hydroxyl species are referred to as Cu-OH modes. (b) The enlarged view, for Cu-OH vibrations, is marked by a dashed box in (a). Vibration bands of the (c) intramolecular $\text{C}\equiv\text{O}$ intermediate and (d) CH_x intermediates.

1. In the manuscript, the authors try to eliminate the undesired oxidative species by applying cathodic potential. For instance, in line 101, “To preclude the effect of a trace amount of undesired oxidative species, such as residual O_2 in the electrolyte or at the electrode surface, we implemented a long-time reduction with Cu electrode in CO_2 saturated KHCO_3 solution at -0.3 V_{RHE} for 5 min.” and in line 112, “To eliminate the pre-introduction of any oxidative groups, a 20 min of long time reduction on the Cu electrode at -0.3 V_{RHE} was performed in CO_2 saturated 0.5 M KHCO_3 containing 100 mM DMPO.”. The authors should unify the conditions for eliminating oxidative species. Most importantly, the authors should prove that they chose the right condition for eliminating any oxidative species in the electrolyte.

Responses 2:

- Thanks very much for your suggestion. To keep the same conditions, 20 min electrochemical reduction was applied. The optical-microscopic images were re-measured, as shown in Figure R2. The same conclusion is achieved. The re-measured result is shown in Figure S7 on page S8 in the *Supporting Information* and correspondingly the revision has been finished on page 5 in the main text.
- In this study, -0.3 V_{RHE} was selected to eliminate the oxidative species in the solutions. When we did not detect any oxidative species except for H^\bullet radical in the electrolyte via the EPR test (Figure 2a), it is the

right condition because the newly generated OH^\bullet radical can be recorded and confirmed. We agree that a longer reduction time than 20 min or more negative potential than $-0.3 \text{ V}_{\text{RHE}}$ is acceptable, and the conditions do not influence the conclusions.

Figure R2. Time-dependent optical-microscopic images of OD-Cu electrodes after switching the potential from $-0.3 \text{ V}_{\text{RHE}}$ (20 min) to OCP.

2. The authors carried out the control experiments in both pure water and K_2SO_4 to show that HCO_3^- is the key for generating OH^\bullet radicals. However, I consider that it is much crucial to show that Cu did not oxidize in these conditions. For instance, the authors should carry out the in-situ Raman experiments mentioned in Fig. 1 for the conditions regarding in line 130-136.

Responses 3:

- Thanks very much for this suggestion. It is a good idea to provide *in-situ* Raman to show the reoxidation processes as shown in Fig. 1. As suggested, we supplemented *in-situ* Raman in the K_2SO_4 electrolyte, and no discernable CuO_x species was detected at OCP after the reduction of Cu_2O to metallic Cu at $-0.3 \text{ V}_{\text{RHE}}$ (Figure R3). It coincides with the EPR result that no OH^\bullet radicals were formed in the K_2SO_4 solution. The supplementary Raman is shown in Figure S6 on page S7 in the *Supporting Information* and correspondingly the discussion has been added on page 4 and page 7 in the main text as follows:

“This rapid reoxidation phenomenon indicates a strongly oxidative species existing in the CO_2 -saturated KHCO_3 electrolyte, in contrast to the non-reoxidation process of OD-Cu in the Ar-saturated 0.25 M K_2SO_4 electrolyte (Supplementary Fig. 6).” (page 4)

“Meanwhile, the reoxidation of surface OD-Cu in 0.25 M K_2SO_4 electrolytes was not observed via in-situ Raman (Supplementary Fig. 6).” (page 7)

- Since the reduction of Cu_2O to metallic Cu at $-0.3 \text{ V}_{\text{RHE}}$ cannot be implemented in pure water owing to the ultrahigh solution resistance, we cannot study the reoxidation process through *in-situ* Raman. Instead, compared to that in the HCO_3^- solution, we have carried out Cu oxidation experiments in the pure water and K_2SO_4 solutions for 24 h, respectively. We also provided the SEM, EPR, EDS, optical microscopic images, and XPS evidence (Fig. 4, Supplementary Figs. 15, 16, 25 and 26).

Figure R3. Real-time Raman spectra of surface Cu_2O species at $-0.3 V_{\text{RHE}}$ and subsequently at OCP in Ar-saturated $0.25 \text{ M K}_2\text{SO}_4$ solution.

3. Regarding the content mentioned in line 164 to line 167, “To further ascertain Cu oxidation associated with electrolyte-induced OH^- , the control experiments were carried out in KHCO_3 solution containing VC, pure water, and K_2SO_4 solution, respectively. No obvious color changes were visually seen for the three cases (insets in Fig. 3 d-f), compared to the pristine Cu plate (inset 167 in Fig. 3 a)” by visually observing the samples is not a reliable method to ascertain the claim. The authors should deploy Raman spectroscopy to investigate the samples since the Raman technique is surface sensitive.

Responses 4:

- Thanks very much for this suggestion. To demonstrate the oxidation processes in these solutions, macroscopically we first have provided visual color changes via the photographs (Insets in Fig. 4a-f) and optical microscope (Supplementary Fig. 15); then microscopically we have analyzed the surface morphology (Fig. 4a-f) and the thickness of corrosion layers (Supplementary Fig. 16) via SEM and analyzed the surface compositions via EDS (Supplementary Figs. 19-25) and XPS (Supplementary Fig. 26).
- Through *in-situ* Raman spectra, we studied the reoxidation behaviors of OD-Cu electrodes in K_2SO_4 solution (Figure R3) and KHCO_3 solution containing VC (Figure R4). No reoxidation phenomena were observed in both solutions after reductions at $-0.3 V_{\text{RHE}}$.

Figure R4. Real-time Raman spectra of surface Cu_2O species at $-0.3 V_{\text{RHE}}$ and subsequently at OCP in CO_2 -saturated 0.5 M KHCO_3 solution containing 10 mM VC .

4. According to the authors' Raman results, it is metallic Cu conducting the CO_2RR reaction. There were no evidence in this research showing that the oxide species can retain during the reaction. As a result, I do not consider that the authors can suggest the presence of $\text{Cu}^{\delta+}$ species during CO_2RR .

Responses 5:

- Thanks very much for this suggestion. As discussed in **Responses 1**, we proposed that the chemical state of Cu and/or the phase of surface Cu species in CO_2RR are the results of dynamic equilibrium between the cathodic reduction and the reoxidation caused by strongly oxidative OH^\bullet radicals in KHCO_3 electrolytes. We provide evidence of what could be the oxidative species leading to the formation of the $\text{Cu}^{\delta+}$ species reported in previous works. As we mentioned in the introduction, "thermodynamically, CuO_x phases should be removed under the CO_2RR conditions thereby the loss of the active $\text{Cu}^{\delta+}$ species, but the $\text{Cu}^{\delta+}$ species still can be frequently observed". We thus propose that the OH^\bullet radical has the chance to get one electron from the Cu catalyst surface.
- At $-0.3 V_{\text{RHE}}$, the cathodic reduction was more dominant than the reoxidation reaction, thus the ultimate chemical state of surface Cu species is mainly metallic Cu. However, it does not mean that the reoxidation reaction does not exist during this process. For example, the Cu-OH vibration still can be observed via *in-situ* Raman at $-0.3 V_{\text{RHE}}$; At $-0.1 V_{\text{RHE}}$ which is much lower than the reduction potential of Cu_2O to metallic Cu ($0.45 V_{\text{RHE}}$), we still clearly observed the Cu-O vibration via *in-situ* Raman (Figure R1).
- We chose $-0.3 V_{\text{RHE}}$ as the study potential because we found that the reduction rate of Cu_2O was moderate for *in-situ* Raman tests. In addition, at this potential, the CO_2RR has taken place, and this potential can avoid the formation of violent gas bubbles.

5. Since the authors try to understand whether the oxide species of oxide-derived copper can retain during CO₂RR, the author should carry out the experiments by using “oxide-derived copper catalyst” instead of Cu plate. Reference such as, J. Am. Chem. Soc. 2012, 134, 17, 7231–7234. would make the result more persuasive.

Responses 6:

- Thanks very much for this suggestion. This work used the term ‘OD-Cu’ because we found that the surface of the Cu-mesh electrode had been oxidized to Cu₂O due to the exposure to air after electrodeposition and to the HCO₃⁻ electrolyte before applying potentials for CO₂RR. The surface Raman test and the further supplementary HRTEM (Figure R5) demonstrate the highly oxidized state. The XRD result may lead to a misunderstanding due to the interference of the Cu-mesh matrix.
- In this work, we have noticed the work from the mentioned reference (J. Am. Chem. Soc. 2012, 134, 17, 7231–7234, ref. 4 in the main text). However, we did not use that method to prepare OD-Cu, because that method cannot produce the electrodes with a powerful surface Raman enhancement effect. Instead, we used the electrodeposition method to deposit the Cu micro-nano particles onto the surface of the Cu-mesh substrate.

Figure R5. HRTEM of as-prepared OD-Cu electrode.

Reviewer #2 (Remarks to the Author):

S. Mu et al. examine experimentally the oxidation of Cu electrocatalyst in CO₂-saturated aqueous electrolytes. By using a trapping strategy, they provide evidence of the formation of OH[•] radicals, to which they attribute the re-oxidation of Cu at the open circuit potential (OCP).

Overall, the work is an important and significant contribution to the study of Cu electrocatalyst employed for CO₂ reduction. As the authors accurately state in the Introduction, the prior literature has what might appear to be conflicting reports ranging from complete reduction of Cu oxides under CO₂RR conditions vs. evidence that some oxides, presumably metastable, persist at relatively negative potentials. In this context, authors provide a plausible mechanism for the formation of such oxide layers. Moreover, they posit that there might be a “see-saw” effect in which there is an interplay between the rate of the reduction reactions and the formation of the oxidizing OH[•] radicals.

Technically, the work is excellent, with impressive and thorough characterization. Raman spectroscopy clearly show the re-oxidation (Figure 1) and I found the use of isotopic labeling to be elegant in supporting the peak assignments (Figure 4). The scavenging experiments using vitamin C are convincing in implication OH radical in the re-oxidation (dark oxidation experiments are also convincing, inset of Figure 3b)

Responses 1:

We truly appreciate your encouraging comments and evaluation of the novelty of our current results.

I support publication of this ms. subject to the following minor revisions.

Authors must discuss the thermodynamic plausibility of the carbonate/water “interplay” (line 230) that produces the OH[•] radicals, even if they do not have a calculation to support their reasoning. Indeed, as the authors state in the Introduction, “CuOx phases should be removed under the CO₂RR operations thereby [causing] the loss of the active Cu^{δ+} species.” Thus an observation that would seem to challenge this thermodynamic view requires some additional comment and, if possible, an explanation. A similar comment applies to the observation of H[•] radicals in the bicarbonate electrolyte solution, page 5.

Responses 2:

- Thanks very much for this suggestion. We supplemented more experiments to investigate the excitation source for the generation of OH[•] radicals in KHCO₃ aqueous solutions. We found that temperature is a key parameter. The room temperature (~25 °C) is enough for the formation of OH[•] radicals (Figure R6). The supplementary EPR is shown in Fig. 3 on page 8 in the main text and correspondingly the discussion has been provided on page 7 in the main text as follows:

“In general, the production of radicals requires excitation sources, yet the signal of DMPO-OH adduct has been tracked at ambient conditions. In this case, the most probable excitation source should be temperature. We hypothesize that the room temperature may activate HCO₃⁻ solutions to produce OH[•] radicals. To confirm it, we implemented the temperature-dependent EPR measurements from 5 to 40 °C in Ar-saturated 0.5 M KHCO₃ electrolyte containing 100 mM DMPO (Fig. 3). When the temperature is as low as 5 °C no EPR signals were detected. An obvious EPR signal from DMPO-OH was observed around 20 °C and increased with the enhanced temperature. This result indicates that temperature is a key parameter for OH[•] radical generation in the KHCO₃ solutions.”

Figure R6. Temperature-dependent EPR spectra in 0.5 M KHCO_3 solutions containing 100 mM DMPO.

Figure R7. Relationships among HCO_3^- concentrations, OH^\bullet radical amount, and reoxidation dynamics of OD-Cu electrodes. **a** The acquired EPR spectra in different concentrations of HCO_3^- solutions containing 100 mM DMPO. **b** Correspondingly, the normalized intensity of DMPO-OH versus HCO_3^- concentration. **c-d** Real-time Raman spectra of OD-Cu reoxidation at OCP after reduction at $-0.3 \text{ V}_{\text{RHE}}$ in different concentrations of KHCO_3 .

- Further, considering that the HCO_3^- aqueous solution itself is crucial for the generation of OH^\bullet radicals, we supplemented experiments to investigate the relationships between HCO_3^- concentration, the intensity of OH^\bullet radicals, and reoxidation dynamics of OD-Cu electrodes (Figure R7). We found that the HCO_3^- contributes to the generation of OH^\bullet radicals that oxidize the Cu surfaces. At $< 0.1 \text{ M}$, there is a linear

relationship between the intensity of DMPO-OH and HCO_3^- concentration. At > 0.1 M, the intensity of DMPO-OH decreases. Thus, HCO_3^- should not be the only factor for the generation of OH^\bullet radicals since there is no linear relationship between HCO_3^- concentration and the intensity of DMPO-OH within the whole HCO_3^- concentration range. For example, cation K^+ might be considered as well when the KHCO_3 concentration is high. Thus, we proposed that the OH^\bullet radicals in HCO_3^- electrolytes were generated under the excitation of room temperature while the dynamic oxygen exchange between HCO_3^- and H_2O is important. The supplementary data is shown in Fig. 5 on page 11 in the main text and correspondingly the discussion has been made on pages 10 and 13 in the main text as follows:

“According to the above analysis, HCO_3^- anions play a vital role in determining the OH^\bullet generation. We thus studied the relationships between HCO_3^- concentration, the intensity of OH^\bullet radicals, and reoxidation dynamics on OD-Cu electrodes. Firstly, we investigated the influence of HCO_3^- concentrations on the amount of OH^\bullet radicals. As shown in Fig. 5a, at < 0.1 M, the DMPO-OH signal increases with increasing the HCO_3^- concentrations, yet further increasing the concentrations cannot produce more OH^\bullet radicals. The optimal HCO_3^- concentration for the formation of OH^\bullet radicals is around 0.1 M (Fig. 5b).

Then, to correlate the OH^\bullet radicals with reoxidation behaviors of OD-Cu electrodes, we carried out in-situ Raman to study the reoxidation phenomena in different concentrations of HCO_3^- solutions. As shown in Fig. 5c-e, with increasing the HCO_3^- concentration from 0.1 to 3.0 M, the Raman intensity of Cu_2O species decreases, and the reappearance of Cu_2O phase delays, during the reoxidation processes after reduction at $-0.3 V_{\text{RHE}}$. Thus, it is indeed that the intensity of DMPO-OH is very related to the reoxidation ability. These results suggest that HCO_3^- contributes to the generation of OH^\bullet radicals that oxidize the Cu surfaces. However, HCO_3^- is not the only factor for the generation of OH^\bullet radicals since there is no linear relationship between HCO_3^- concentration and the intensity of DMPO-OH.” (page 10)

“Thus, we show that the OH^\bullet radicals have been generated through excitation of room temperature in HCO_3^- electrolytes while the dynamic oxygen exchange between HCO_3^- and H_2O supplies oxygen source for the formation of OH^\bullet radicals.” (page 13)

- The H^\bullet radicals are generated via the step $\text{H}^+ + \text{e}^- \rightarrow \text{H}^\bullet$ in HER. The discussion has been added on page 5 in the main text as follows:

“At $-0.3V_{\text{RHE}}$, the hydrogen evolution reaction (HER) occurs and the generated hydrogen radicals (H^\bullet) can be trapped as a DMPO-H adduct (hyperfine splitting constants, $A_{\text{N}} = 1.65$ mT, $A_{\text{H}} = 2.25$ mT)”

- We further discussed the ‘reduction hysteresis’ of Cu_2O to metallic Cu. The discussion has been added on page 4 in the main text as follows:

“We found that the surface Cu species go through the process: $\text{Cu}_2\text{O} \rightarrow \text{CuO}_x \rightarrow$ metallic Cu, with the cathode potential decreasing from OCP to (Supplementary Fig. 2). It is a reverse process when switching from $-0.3 V_{\text{RHE}}$ to OCP. Thus, we suggest that the chemical state of Cu and the phase of surface Cu species are the results of dynamic equilibrium between the cathodic reduction and the reoxidation caused by a sort of strongly oxidative species in KHCO_3 electrolytes.”

In the same vein, is the presence of Cu (or a metal surface) required to generate the radicals? That is, the experiments described in lines 108-129 have a Cu electrode presence. Would radicals be made in a similar amount if the metal were not present?

Responses 3:

- Thanks very much for this question. The presence of a Cu electrode is not necessary. The electrochemical reduction on the Cu electrode is to remove the pre-existing oxidative substances in the electrolytes. If the metal electrode was not present, a similar amount of OH radicals were produced. For example, the experiments of *Supplementary Fig. 9* (here left in Figure R8) and *Supplementary Fig. 11* (here right in Figure R8).

Figure R8. (Left) Time-dependent EPR spectra of the CO₂-saturated 0.5 M KHCO₃ solution containing 100 mM DMPO. (Right) EPR spectra of the KHCO₃/K₂SO₄ mixed solutions. The solutions at different HCO₃⁻/SO₄²⁻ mole ratios contain 100 mM DMPO under the same K⁺ concentrations of 0.5 M.

Minor revisions, not technical in nature.

In Figure 2 (and in most of the EPR spectra), the fits shown in color are more prominent than the data, which is shown with a thin line in light grey. The figure should be modified such that the experimental data is more prominent than the fit.

Responses 4:

Thanks very much for this suggestion. We have modified the EPR spectra and now the experimental data is prominent.

Line 179, replace “corrosives” with “corrosion products.”

Responses 5:

We have replaced “corrosives” with “corrosion products.”.

Line 147, replace “OH adduct disappeared, instead of a” with OH adduct disappeared and was replaced by”

Responses 6:

We have replaced “OH adduct disappeared, instead of a” with “OH adduct disappeared and was replaced by”.

Reviewer #3 (Remarks to the Author):

The major claim of this manuscript is that during CO₂RR, HCO₃⁻ and H₂O reacts to give OH[•] radicals. These OH[•] radicals are proposed to oxidize the Cu surface during CO₂RR. Experimental evidence presented include the use of in situ Raman spectroscopy, secondary ion mass spectrometry, and isotope-labelling.

Overall, I do not agree with the claims of this study, as presented in the manuscript. There are major gaps that need to be addressed, before this manuscript can be re-considered.

Responses 1:

We highly appreciate the suggestions to improve the quality of this paper.

Major:

Authors claimed that HCO₃⁻ and H₂O reacts to give OH[•] radicals. And the OH[•] radicals are responsible for oxidizing the Cu surface. But this does not make sense, since in other electrolytes such as KOH (without any HCO₃⁻ present), the Cu surfaces will also quickly oxidize once cathodic potentials are removed.

Responses 2:

- Thanks very much for this question. We agree with the referee that the Cu surfaces may be quickly oxidized in the other solutions. Nevertheless, in this work, we focus on “why Cu^{δ+} species can be frequently observed in HCO₃⁻ solutions in previous reports”, and “what could be the oxidative species”.
- We used the *in-situ* Raman spectra to study the reoxidation behaviors of OD-Cu in the KHCO₃ solution containing VC and the K₂SO₄ solution after the reduction of surface Cu₂O to metallic Cu at -0.3 V_{RHE} (Figure R9) respectively, where no OH[•] radicals in both solutions. We found that there are no reoxidation phenomena taking place in both solutions. The supplementary Raman data of the K₂SO₄ solution is shown in Supplementary Fig. 6 on page S7 in the *Supporting Information* and correspondingly the discussion has been added on page 4 and page 7 in the main text as follows:

“This rapid reoxidation phenomenon indicates a strongly oxidative species existing in the CO₂-saturated KHCO₃ electrolyte, in contrast to the non-reoxidation process of OD-Cu in the Ar-saturated 0.25 M K₂SO₄ electrolyte (Supplementary Fig. 6).” (page 4)

“Meanwhile, the reoxidation of surface OD-Cu in 0.25 M K₂SO₄ electrolytes was not observed via in-situ Raman test (Supplementary Fig. 6).” (page 7)

Figure R9. Studies of the reoxidation behaviors of OD-Cu in the KHCO_3 solution containing VC (**Left**) and the K_2SO_4 solution (**Right**) after the reduction at $-0.3 \text{ V}_{\text{RHE}}$, via *in-situ* Raman spectra.

- We carried out more experiments to study the excitation source for the generation of OH^\bullet radicals in KHCO_3 aqueous solutions. We found that temperature is a key parameter. The room temperature ($\sim 25^\circ\text{C}$) is enough for the formation of OH^\bullet radicals (Figure R10). The supplementary EPR is shown in Fig. 3 on page 8 in the main text and correspondingly the discussion has been provided on page 7 in the main text as follows:

“In general, the production of radicals requires excitation sources, yet the signal of DMPO-OH adduct has been tracked at ambient conditions. In this case, the most probable excitation source should be temperature. We hypothesize that the room temperature may activate HCO_3^- solutions to produce OH^\bullet radicals. To confirm it, we implemented the temperature-dependent EPR measurements from 5 to 40 °C in Ar-saturated 0.5 M KHCO_3 electrolyte containing 100 mM DMPO (Fig. 3). When the temperature is as low as 5 °C no EPR signals were detected. An obvious EPR signal from DMPO-OH was observed around 20 °C and increased with the enhanced temperature. This result indicates that temperature is a key parameter for OH^\bullet radical generation in the KHCO_3 solutions.”

Figure R10. Temperature-dependent EPR spectra in 0.5 M KHCO_3 solutions containing 100 mM DMPO.

- Pg 5- Are the authors suggesting that HER occurs at OCP on Cu? How can this be? What is the OCP? Is H₂ gas detected? Can the authors use gas chromatography to verify this?

Responses 3:

- We are sorry for this misunderstanding. HER cannot happen at OCP. For the measurements of radicals, considering the very short lifetimes of radicals (up to several ms), DMPO as a spin trapping agent was added into KHCO₃ electrolytes, otherwise, we cannot detect any radicals in solutions via EPR. Therefore, DMPO-H radical adduct generated during the HER did not disappear as soon as we stopped the bias, owing to its increased lifetime, thus DMPO-H radical adduct can be measured even if we switched the potential from -0.3 V_{RHE} to OCP. The further discussion has been added on page 5 in the main text as follows:

“Considering the very short lifetimes of radicals (up to several ms), 5,5-dimethyl-1-pyrroline N-oxide (DMPO) as a spin trapping agent was added into KHCO₃ electrolytes, thus the formed DMPO-radical adducts have the lifetimes as long as tens of minutes³⁸, facilitating the EPR tests.”

“At -0.3V_{RHE}, the hydrogen evolution reaction (HER) occurs, and the generated hydrogen radicals (H[•]), formed via H⁺ + e⁻ → H[•], can be trapped as a DMPO-H adduct (hyperfine splitting constants, A_N = 1.65 mT, A_H = 2.25 mT).”

“As shown in Fig. 2a, DMPO-H radical adduct generated during the HER did not disappear as soon as we stopped the bias, owing to the increased lifetime, thus they can be measured even if we switched the potential from -0.3 V_{RHE} to OCP.”

- OCP represents open circuit potential, which has been provided on page 3 and in Fig. 1 on page 4, in the main text.
- After stopping the applied bias, no HER takes place, so no H₂ can be detected by gas chromatography.

- What is the lifetime and stability of the OH[•] radical? How can it be so long that its signal keeps increasing (Suppl Fig 5)?

Responses 4:

- Thanks very much for this question. The lifetime of OH[•] radicals is very short (up to several ms), which is far below the time-resolution of EPR. Therefore, we added DMPO to trap OH[•] radicals via the formation of DMPO-OH adducts, whose lifetime is much longer.
- We agree with the referee that the EPR intensity of DMPO-OH will decrease with increasing time if there are no newly generated OH[•] radicals. In this case, OH[•] radicals from HCO₃⁻ solutions are produced all the time under the excitation of room temperature while the concentration of DMPO is enough (100 mM), thus the newly formed OH[•] radicals can be trapped by DMPO, leading to an increase of DMPO-OH concentration thereby increasing the EPR signal.

- Could a HCO₃⁻ concentration dependent studies be done? I will assume that if the authors are correct, a higher HCO₃⁻ concentration will lead to faster oxidation of the Cu surface.

Responses 4:

- Thanks very much for this question. We supplemented more experiments to investigate the relationships between HCO₃⁻ concentration, the intensity of OH[•] radicals, and the reoxidation dynamics of OD-Cu

electrodes (Figure R11). We found that the HCO_3^- contributes to the generation of OH^\bullet radicals that oxidize the Cu surfaces. At < 0.1 M, there is a linear relationship between the intensity of DMPO-OH and HCO_3^- concentration. At > 0.1 M, the intensity of DMPO-OH decreases. Thus, HCO_3^- should not be the only factor for the generation of OH^\bullet radicals since there is no linear relationship between HCO_3^- concentration and the intensity of DMPO-OH within the whole HCO_3^- concentration range. For example, cation K^+ might be considered as well when the KHCO_3 concentration is high. Thus, we proposed that the OH^\bullet radicals in HCO_3^- electrolytes were generated under the excitation of room temperature while the dynamic oxygen exchange between HCO_3^- and H_2O is important. The supplementary data is shown in Fig. 5 on page 11 in the main text and correspondingly the discussion has been made on pages 10 and 13 in the main text as follows:

“According to the above analysis, HCO_3^- anions play a vital role in determining the OH^\bullet generation. We thus studied the relationships between HCO_3^- concentration, the intensity of OH^\bullet radicals, and reoxidation dynamics on OD-Cu electrodes. Firstly, we investigated the influence of HCO_3^- concentrations on the amount of OH^\bullet radicals. As shown in Fig. 5a, at < 0.1 M, the DMPO-OH signal increases with increasing the HCO_3^- concentrations, yet further increasing the concentrations cannot produce more OH^\bullet radicals. The optimal HCO_3^- concentration for the formation of OH^\bullet radicals is around 0.1 M (Fig. 5b).

Then, to correlate the OH^\bullet radicals with reoxidation behaviors of OD-Cu electrodes, we carried out in-situ Raman to study the reoxidation phenomena in different concentrations of HCO_3^- solutions. As shown in Fig. 5c-e, with increasing the HCO_3^- concentration from 0.1 to 3.0 M, the Raman intensity of Cu_2O species decreases, and the reappearance of Cu_2O phase delays, during the reoxidation processes after reduction at -0.3 V_{RHE}. Thus, it is indeed that the intensity of DMPO-OH is very related to the reoxidation ability. These results suggest that HCO_3^- contributes to the generation of OH^\bullet radicals that oxidize the Cu surfaces. However, HCO_3^- is not the only factor for the generation of OH^\bullet radicals since there is no linear relationship between HCO_3^- concentration and the intensity of DMPO-OH.” (page 10)

Figure R11. Relationships among HCO_3^- concentrations, OH^\bullet radical amount, and reoxidation dynamics of OD-Cu electrodes. **a** The acquired EPR spectra in different concentrations of HCO_3^- solutions containing 100 mM DMPO. **b** Correspondingly, the normalized intensity of DMPO-OH versus HCO_3^- concentration. **c-d** Real-time Raman spectra of OD-Cu reoxidation at OCP after reduction at $-0.3 V_{\text{RHE}}$ in different concentrations of KHCO_3 .

- What is the mechanism for OH^\bullet radical formation from $\text{HCO}_3^- + \text{H}_2\text{O}$?

Responses 5:

- Thanks very much for this question. We supplemented more experiments to investigate the excitation source for the generation of OH^\bullet radicals in KHCO_3 aqueous solutions. We found that temperature is a key parameter. For instance, the low temperature of 5 °C cannot cause the generation of OH^\bullet radicals, but the room temperature (~25 °C) is enough for the formation of OH^\bullet radicals (Figure R10). The supplementary EPR is shown in Fig. 3 on page 8 in the main text and correspondingly the discussion has been provided on page 7 in the main text as follows:

“In general, the production of radicals requires excitation sources, yet the signal of DMPO-OH adduct has been tracked at ambient conditions. In this case, the most probable excitation source should be temperature. We hypothesize that the room temperature may activate HCO_3^- solutions to produce OH^\bullet radicals. To confirm it, we implemented the temperature-dependent EPR measurements from 5 to 40 °C in Ar-saturated 0.5 M KHCO_3 electrolyte containing 100 mM DMPO (Fig. 3). When the temperature is as low as 5 °C no EPR signals were detected. An obvious EPR signal from DMPO-OH was observed around 20 °C and increased with the enhanced temperature. This result indicates that temperature is a key parameter for OH^\bullet radical generation in the KHCO_3 solutions.”

- We supplemented experiments to investigate the relationships among HCO_3^- concentrations, intensities of OH^\bullet radicals and reoxidation dynamics of OD-Cu electrodes (Figure R11). We found that the HCO_3^- contributes to the generation of OH^\bullet radicals that further oxidize the Cu surfaces. However, HCO_3^- is not the only factor for the generation of OH^\bullet radicals. The solvent H_2O , rather than pure H_2O solution, also plays a role. Therefore, based on our current experimental results, we propose that the OH^\bullet radicals have been generated through the excitation of room temperature in HCO_3^- electrolytes while the dynamic oxygen exchange between HCO_3^- and H_2O supplies an oxygen source for the formation of OH^\bullet radicals.

- End of pg 6 and start of Pg 7: The authors do not realize that water can also serve as a source of O to oxidize Cu. It does not need to be O_2 gas or OH^\bullet radical (if these do exist).

Responses 6:

- Thanks very much for this comment. We would take the view that even if pure H_2O is able to oxidize Cu, it should be at a much slower rate. Within a reasonable period, for instance, within 24 h, H_2O cannot cause obvious oxidation of the Cu plate, unlike the one with deep oxidation in HCO_3^- aqueous solution (Fig. 4 in the main text).
- Based on the *in-situ* Raman, the oxidation of Cu to Cu_2O can take place within 60 s. It is a fast process. Within 24 h, the thickness of the oxide layer could be around 500 nm.

Minor:

- In the introduction of the manuscript, the authors wrote statements such as ‘This is due to the partially charged $\text{Cu}^{\delta+}$ species that plays a crucial role during CO_2RR ’, ‘ $\text{Cu}^{\delta+}$ species has been frequently detected

during CO₂RR.[?], etc. I think this is an unbalanced introduction. The existence of Cu^{δ+} species during CO₂RR is still controversial. Papers that claim that this species exist often based their claim on results obtained from ex-situ methods. Since Cu oxidizes so easily, the data obtained from ex-situ methods are questionable. I suggest that the authors try to balance out their discussion.

Responses 7:

- Thanks very much for this suggestion. We replaced the previous statements with “This is likely due to the partially charged Cu^{δ+} species that plays a crucial role”.
- We agree with your view that it has not reached a consensus on whether Cu^{δ+} species can exist at catalyst surfaces or be stable under the cathodic electrolysis in CO₂RR. We supplemented a discussion about this topic. It has been provided on page 2 in the main text as follows:

“Thermodynamically, CuO_x phases should be removed under the CO₂RR operations thereby the loss of the active Cu^{δ+} species¹⁵. Some studies have demonstrated the reduction of CuO_x phases to metallic Cu during CO₂RR¹⁶⁻²⁰. Interestingly, despite these, the Cu^{δ+} species has been frequently detected in CO₂RR^{8,9,21-23}.”

- We have provided the cited works that use *in-situ* characterization methods to demonstrate that Cu^{δ+} species in CO₂RR, such as Ref. 8, 9, 21, 24, and 26.

- Fig 1: I suggest that authors show their spectra to at least 2500 cm⁻¹ so that the C-O stretching vibration of CO(ad) at about 2000 cm⁻¹ can be seen.

Responses 8:

- Thanks very much for this suggestion. To focus on the oxidation of Cu, we did not provide the wide Raman spectra in the previous version. We supplemented the wide Raman spectra and detected the vibrations of C≡O at 2070-2100 cm⁻¹ and the vibrations of CH_x at 2820-2950 cm⁻¹, from 0.2 to -0.6 V_{RHE} (Figure R12). The supplementary Raman data were shown in Supplementary Fig. 3 on page S4 and Supplementary Fig. 4 on page S5 in the *Supporting Information* and correspondingly the discussion has been provided on page 3 in the main text as follows:

“In addition, the peaks at 282, 360, 2070-2100 cm⁻¹ are related to the frustrated ρ (Cu-C-O) rotational mode, ν (Cu-CO) stretching mode, and intramolecular C≡O stretching vibration of CO intermediates respectively. The bands at 2820-2950 cm⁻¹ are assigned to the -CH_x stretching regions from 0.2 to -0.6 V_{RHE} (Fig. 1a and b, Supplementary Figs. 2-4)^{30,31}, demonstrating the occurrence of CO₂RR.”

Figure R12. Raman vibration bands of the intramolecular C≡O intermediate and CH_x intermediates during the CO₂RR for OD-Cu electrodes in CO₂-saturated 0.5 M KHCO₃ solution at different indicated potentials.

REVIEWER COMMENTS

Reviewer #1 (Remarks to the Author):

The content of the revised manuscript has greatly improved, yet there are some issues needed to be addressed.

1. On page 3, line 79, “. The bands at 2820 2950 cm^{-1} are assigned to the CH_x stretching regions regions^{30,31} from 0.2 to -0.6 VRHE (Fig. 1a and b, Supplementary Figs. 2-4), demonstrating the occurrence of CO₂RR.”, to my knowledge, the onset of hydrocarbon products in a typical CO₂RR in 0.5 M KHCO₃ without using a flow cell system is \sim -0.5 V. Why the authors can obtain CH_x stretching regions even at 0.2 V vs. RHE. Moreover, it is unlikely that we can get a cathodic current at 0.2 V vs. RHE. Besides, the authors may provide the current vs t profile in the supporting information.
2. Why did the authors choose K₂SO₄ as a controlled measurement? The authors should provide a robust explanation in the manuscript.
3. The authors should provide experimental setups of in situ Raman and EPR.
4. On page 8 line 190 “To preclude the CO₂(aq) effect, the same operation was implemented in Ar-saturated KHCO₃, and a darker Cu surface was obtained (inset in Fig. 4 cc). Thus, the Cu oxidation by CO₂(aq) was ruled out.”, However, the pH varies by purging different gas. Can the authors rule out this effect?
5. On page 10 line 239, “However, HCO₃ is not the only factor for the generation of OH radicals since there is no linear relationship between HCO₃ concentration and the intensity of DMPOOH.”, can the authors provide other potential factors for the generation of OH radicals?

Reviewer #2 (Remarks to the Author):

The revised version of “ Hydroxyl Radicals Dominate Reoxidation of Oxide-derived Cu in Electrochemical CO₂ Reduction” fully addresses the comments I had on the original version of the ms. However, some of the additions of the authors have introduced new concerns, which should be addressed in a minor revision.

Abstract. "formed via the excitation of room temperature in HCO₃⁻ solutions." While the thermal energy in the water could be viewed as an excitation source I suggest the use of more scientifically precise language here and throughout the ms.: formed *at room temperature*

Introduction. "We took the view that the Cu δ⁺ species should be dynamically existing," Does this mean that this species is formed and also reacts back to Cu(0) at some rate?

Page 3. Line 57. "we observed serious oxidizing corrosion" What criterion was used to assess "serious" as opposed to lesser degrees of corrosion?

Page 7, lines 173-181. The authors should strive to use more chemically correct language. I believe that they consider whether OH radical formation could be a thermally activated process motivating them to do the temperature dependent study.

Figure 3 should have "C" next to the numbers for the 4 traces to indicate temperature. Also, can authors rule out the possibility of a temperature dependence for the efficiency of the DMPO scavenging?

Page 10, lines 230-231. "yet further increasing the concentrations cannot produce more OH•radicals" Actually, the OH radical concentration seems to go down, not saturate. Can the authors comment on this finding?

Page 10, lines 238-241 and Figure 5b. There is not a linear relationship between HCO₃ and DMPO-OH (actually, it would not be apparent on the semilog plot used). There are lines on Figure 5b. What is their meaning? If they are guides to the eye, this should be stated.

Line 288. The Discussion section actually appears to be the Conclusion of the study.

Reviewer #3 (Remarks to the Author):

I thank the authors for their revised manuscript and responses to my queries, which are largely ok. I have further queries:

Major:

For Fig R12: Authors claimed to see CO stretching and C-H bands during CO₂RR at 0.2 V RHE and more negative potentials. I don't think this can be correct. It is not possible for CO₂RR to occur at such positive potentials of 0.2 V RHE (goes against the thermodynamics).

Expts for Suppl Fig 7 – how do the authors know that there is no O₂ present?

In experiments using other electrolytes, were OH radicals also found?

Could the authors explain why the formation of OH radicals decreased when concentration of HCO₃ increased beyond 0.1 M?

Minor:

In the Introduction, the authors stated in the 4th-5th line that partially-charged Cu^{δ+} species is likely to play a crucial role. I find that this is a misleading statement. As mentioned in my earlier review, this is a statement that has not been fully verified.

Dear Reviewers:

We highly appreciate your constructive suggestions to improve this manuscript. Followed by the suggestions from the second round, we have further carried out experiments and carefully revised the manuscript. Corresponding responses and revisions have been made in a point-by-point manner as follows. The revisions are marked in blue the revised manuscript for your convenience.

Reviewer #1 (Remarks to the Author):

The content of the revised manuscript has greatly improved, yet there are some issues needed to be addressed.

We highly appreciate the positive comment and previous professional suggestions from the reviewer. The comments are quite helpful, please find our response and revision as follows.

1. On page 3, line 79, “. The bands at 2820 2950 cm^{-1} are assigned to the CH_x stretching regions 30,31 from 0.2 to -0.6 VRHE (Fig. 1a and b, Supplementary Figs. 2-4), demonstrating the occurrence of CO_2RR .”, to my knowledge, the onset of hydrocarbon products in a typical CO_2RR in 0.5 M KHCO_3 without using a flow cell system is ~ -0.5 V. Why the authors can obtain CH_x stretching regions even at 0.2 V vs. RHE. Moreover, it is unlikely that we can get a cathodic current at 0.2 V vs. RHE. Besides, the authors may provide the current vs t profile in the supporting information.

Responses 1:

- We are thankful to the reviewer for pointing out this concern. We completely agree with the referee that, without using a flow cell, the generation of hydrocarbon products is usually reported below $-0.5 V_{\text{RHE}}$, based on gas chromatography and NMR tests. However, the high detection limit of these techniques makes it difficult to probe the reaction initiation point, where the surface binding intermediates are already present, yet the desorption and/or production of final products is not significant. Luckily, with *in-situ* surface-enhanced Raman spectroscopy, C-H vibration signals from binding intermediates, which is the initiation sign of the reaction, can be detected under more positive potentials.
- Meanwhile, it can also be found in other studies that the CO_2RR intermediates can be detected at the potentials close to 0.2 V_{RHE} , for example, CO_{ad} vibration was observed via *in-situ* Raman at 0.2 V_{RHE} as well (J. Am. Chem. Soc. 2017, 139, 3774-3783); CH_4 was generated at 0.16 V_{RHE} (ACS Appl. Energy Mater 2020, 3, 1, 1119-1127).
- To further verify the cathodic Faradaic current, as suggested, we supplemented the current-time curves for the potential-dependent Raman test (Figure R1). The supplementary result is shown in Supplementary Fig. 2c and 2d on page S4 in Supporting Information.

Figure R1. Current-time curves at the indicated potentials versus RHE for potential-dependent Raman tests in CO₂-saturated 0.5 M KHCO₃ solution. Inset shows an amplified view.

2. Why did the authors choose K₂SO₄ as a controlled measurement? The authors should provide a robust explanation in the manuscript.

Responses 2:

- We thank the reviewer for this kind suggestion. K₂SO₄ was carefully chosen as a control because: (1) it has moderate solubility (~0.6 M) relative to such as KClO₄ (~0.12 M with saturation); (2) SO₄²⁻ anion is relatively stable; (3) SO₄²⁻ has relatively weaker interaction with Cu relative to such as KCl and KI.
- We added this supplementary information into the main text on page 4:

“Here K₂SO₄ was selected as a control electrolyte because of its moderate solubility relative to KClO₄, suitable chemical stability, and relatively weaker interaction with Cu relative to such as KCl and KI.”

3. The authors should provide experimental setups of in situ Raman and EPR.

Responses 3:

- We thank the reviewer for this suggestion, and it is helpful for the readers to quickly understand the experimental methods. The supplementary experimental setup images of *in-situ* Raman have been provided (Figure S2d on page S4 in Supporting Information).
- EPR instrument equipped with the capillary tube has been provided (Figure S8b on page S10 in Supporting Information).

Figure R2. Experimental setup of *in-situ* electrochemical Raman spectroscopy.

Figure R3. EPR instrument equipped with the capillary tube.

4. On page 8 line 190 “To preclude the $\text{CO}_2(\text{aq})$ effect, the same operation was implemented in Ar-saturated KHCO_3 , and a darker Cu surface was obtained (inset in Fig. 4 cc). Thus, the Cu oxidation by $\text{CO}_2(\text{aq})$ was ruled out.” , However, the pH varies by purging different gas. Can the authors rule out this effect?

Responses 4:

- We thank the reviewer for this question. This sentence was to introduce that $\text{CO}_2(\text{aq})$ did not induce the generation of DMPO-OH. We observed that the CO_2 -purged electrolyte even induces a slightly lower intensity of DMPO-OH (Fig. 4 in the main text) relative to that in the Ar-purged electrolyte.
- We agree that based on the $\text{CO}_2/\text{HCO}_3^-$ equilibrium, theoretically, it is challenging to control the pH and $\text{CO}_2(\text{aq})$ concentration simultaneously. We found that the pH value of CO_2 -saturated KHCO_3 is around 7.5, while the pH value of Ar-purged KHCO_3 is about 8.2. When we refill the Ar-saturated KHCO_3 with CO_2 for a short duration, similar pH can be retained and similar results can be observed, thus we conclude that the pH effect is not significant here.
- To improve the manuscript, we rewrote the discussion in the main text on page 8.

“To preclude the $\text{CO}_2(\text{aq})$ effect, the same operation was implemented in Ar-saturated KHCO_3 , and a darker Cu surface was observed associated with stronger DMPO-OH (inset in Fig. 4c).”

5. On page 10 line 239, “However, HCO_3^- is not the only factor for the generation of OH radicals since there is no linear relationship between HCO_3^- concentration and the intensity of DMPO-OH.”, can the authors provide other potential factors for the generation of OH radicals?

Responses 5:

- Thanks very much for this question. We found that the HCO_3^- contributes to the generation of OH^\bullet radicals that oxidize the Cu surfaces. At $< 0.1 \text{ M}$, the intensity of DMPO-OH increases with HCO_3^- concentration. At $> 0.1 \text{ M}$, the intensity of DMPO-OH decreases.
- Upon increasing the KHCO_3 concentration, the K^+ concentration has been increased as well. Thus, we found that, at the same K^+ concentration (0.5 M), the relationship between the HCO_3^- concentration and the intensity of DMPO-OH is more pronounced (Figure R4) following the order $0.2 \text{ M} < 0.4 \text{ M} \leq 0.5 \text{ M}$, and then the intensity of DMPO-OH tends to saturate.
- We supplemented the discussion in the main text on page 10.

“It is worth noting that further increasing the HCO_3^- concentration enhances the K^+ concentration as well. Thus, at the same K^+ concentration (0.5 M) with K^+ compensation by K_2SO_4 , the relationship between the HCO_3^- concentration and the intensity of DMPO-OH is more pronounced following the order $0.2 \text{ M} < 0.4 \text{ M} \leq 0.5 \text{ M}$ (Supplementary Fig. 10), and the intensity of DMPO-OH tends to saturate. The optimal HCO_3^- concentration without K^+ compensation for the OH^\bullet radical formation is around 0.1 M (Fig. 5b).”

Figure R4 (Supplementary Fig. 10). EPR spectra of the $\text{KHCO}_3/\text{K}_2\text{SO}_4$ mixed solutions. The solutions at different $\text{HCO}_3^-/\text{SO}_4^{2-}$ mole ratios contain 100 mM DMPO under the same K^+ concentrations of 0.5 M .

Reviewer #2 (Remarks to the Author):

The revised version of “Hydroxyl Radicals Dominate Reoxidation of Oxide-derived Cu in Electrochemical CO₂ Reduction” fully addresses the comments I had on the original version of the ms. However, some of the additions of the authors have introduced new concerns, which should be addressed in a minor revision.

We thank the reviewer for the positive evaluation of the previously revised manuscript. We highly appreciate the suggestions to improve the quality of this paper. Please find the corresponding response and revision below.

1. Abstract. “formed via the excitation of room temperature in HCO₃⁻ solutions.” While the thermal energy in the water could be viewed as an excitation source I suggest the use of more scientifically precise language here and throughout the ms.: formed *at room temperature*

Responses 1:

- We thank the reviewer for the suggestions on language improvement. Throughout the revised manuscript, “formed via the excitation of room temperature in HCO₃⁻ solutions.” has been revised to “formed at room temperature”.
2. Introduction. “We took the view that the Cu ^{δ+} species should be dynamically existing,” Does this mean that this species is formed and also reacts back to Cu(0) at some rate?

Responses 2:

- We thank the reviewer for the careful reading. Metallic Cu (0) can be oxidized by the OH[•] radicals, while the oxidized Cu species are also reduced back towards Cu (0) under reductive potentials. The observed Cu^{δ+} species are the result of dynamic competition between the mentioned two reactions.
3. Page 3. Line 57. “we observed serious oxidizing corrosion” What criterion was used to assess “serious” as opposed to lesser degrees of corrosion?

Responses 3:

- We thank the reviewer for the detailed questions. This “serious” oxidizing corrosion was used to assess higher degrees of corrosion relative to those electrolytes without OH[•] radicals. The previous description was ambiguous, thus we revised it accordingly.
 - On page 3, the content has been revised into “..., we observed higher degrees of oxidizing corrosion of Cu electrodes in CO₂- or Ar-saturated KHCO₃ solution under open circuit potential (OCP) relative to those electrolytes without OH[•] radicals...”
4. Page 7, lines 173-181. The authors should strive to use more chemically correct language. I believe that they consider whether OH radical formation could be a thermally activated process motivating them to do the temperature dependent study.

Responses 4:

- We sincerely appreciate the reviewer for the patience in language improvement. On page 7, the first sentence to introduce a thermally activated process has been revised into “To consider whether the formation of OH[•] radicals could be a thermally activated process, the signal of DMPO-OH adduct has been tracked via a temperature-dependent study.”

5. Figure 3 should have “C” next to the numbers for the 4 traces to indicate temperature. Also, can authors rule out the possibility of a temperature dependence for the efficiency of the DMPO scavenging?

Responses 5:

- We thank the reviewer for this consideration. We have corrected the temperature unit and made °C next to the numbers.
- In this study, we used 100 mM DMPO (high enough concentration), thus the formation rate of DMPO-OH was proportional to the generation rate of OH radicals. Based on the previous report (Res Chem Intermed, 2012, 38:2191–2204), the trapping rate of DMPO decreases slightly with the reaction temperature. Therefore, the increase of DMPO-OH signal intensity with the reaction temperature is efficient and the trend of the variation of DMPO-OH with temperature remains.

6. Page 10, lines 230-231. “ yet further increasing the concentrations cannot produce more OH•radicals” Actually, the OH radical concentration seems to go down, not saturate. Can the authors comment on this finding?

Responses 6:

- Thanks very much for this question. When we increased the KHCO₃ concentration, both the concentrations of K⁺ and HCO₃⁻ have been increased. However, if we keep the K⁺ at 0.5 M and increase the HCO₃⁻ concentration, the relationship between the HCO₃⁻ concentration and the intensity of DMPO-OH is more pronounced (Figure R5) following the order 0.2 M < 0.4 M ≤ 0.5 M, and then the intensity of DMPO-OH tends to saturate. It means that too high K⁺ concentration is not good for the generation of OH[•] radicals.
- We supplemented the discussion in the main text on page 10.

“It is worth noting that further increasing the HCO₃⁻ concentration enhances the K⁺ concentration as well. Thus, at the same K⁺ concentration (0.5 M) with K⁺ compensation by K₂SO₄, the relationship between the HCO₃⁻ concentration and the intensity of DMPO-OH is more pronounced following the order 0.2 M < 0.4 M ≤ 0.5 M (Supplementary Fig. 10), and the intensity of DMPO-OH tends to saturate. The optimal HCO₃⁻ concentration without K⁺ compensation for the OH[•] radical formation is around 0.1 M (Fig. 5b).”

Figure R5 (Supplementary Fig. 10). EPR spectra of the KHCO₃/K₂SO₄ mixed solutions. The solutions at different HCO₃⁻/SO₄²⁻ mole ratios contain 100 mM DMPO under the same K⁺ concentrations of 0.5 M.

7. Page 10, lines 238-241 and Figure 5b. There is not a linear relationship between HCO_3^- and DMPO-OH (actually, it would not be apparent on the semilog plot used). There are lines on Figure 5b. What is their meaning? If they are guides to the eye, this should be stated.

Responses 7:

- We thank the reviewer for kindly reminding us of the statement for the guiding line. We agree with the referee that there is no linear relationship between HCO_3^- concentration and DMPO-OH intensity. The gray lines in Figure 5b as indicators show the trends that DMPO-OH intensity varies with HCO_3^- concentration.
- The x-axis has been revised; the “linear relationship” relevant information has been deleted to avoid misleading.
- Correspondingly, the illustration has been added in the caption of Figure 5 on page 11.

“The gray lines guide the trends of DMPO-OH intensity without the compensation of K^+ .”

8. Line 288. The Discussion section actually appears to be the Conclusion of the study.

Responses 8:

We thank the reviewer for the careful reading. As suggested, the discussion subtitle has been revised to “Conclusions”

Reviewer #3 (Remarks to the Author):

I thank the authors for their revised manuscript and responses to my queries, which are largely ok. I have further queries:

We sincerely thank the reviewer for the positive comments after the first revision. Please find our latest response and revision for your new suggestions as follows.

Major:

1. For Fig R12: Authors claimed to see CO stretching and C-H bands during CO₂RR at 0.2 V RHE and more negative potentials. I don't think this can be correct. It is not possible for CO₂RR to occur at such positive potentials of 0.2 V RHE (goes against the thermodynamics).

Responses 1:

- We thank the reviewer for the constructive comment. Firstly, we agree with the reviewer that no massive products can be detectable by the gas chromatography and NMR, which are the most common techniques to quantify products, at such potential, owing to their detection limit. However, with surface enhancement, *in-situ* Raman can detect the surface binding intermediates, which is the characteristic of initiation of cathodic reactions (CO₂ has been reduced to intermediates binding on the surface of catalysts), before significant evolution of CO₂RR products.

- The CO₂RR taking place at the potentials close to 0.2 V_{RHE} can be also found in the other works, for example, CO_{ab} vibration was observed via *in-situ* Raman at 0.2 V_{RHE} (J. Am. Chem. Soc. 2017, 139, 3774-3783); CH₄ was generated at 0.16 V_{RHE} (ACS Appl. Energy Mater 2020, 3, 1, 1119-1127); A simulated potential-dependent v-DoS of *HOC-COH at 0.21V (PNAS, 2019, 116, 7718-7722). Thus, it suggests that some □ -CH_x and CO intermediates form at 0.2 V RHE on the surface of Cu catalysts.

2. Expts for Suppl Fig 7 – how do the authors know that there is no O₂ present?

Responses 2:

- We thank the reviewer for the careful consideration of experiment details. Ar and subsequent CO₂ bubbling through the KHCO₃ can remove the existing O₂ in the electrolyte (the protocol as shown in Figure R6). The reaction cell has been purged by Ar gas before introducing electrolyte. Besides, the reaction chamber has been separated by an exchange membrane separated cell (Figure R6). Meanwhile, we implemented an electrochemical reduction at -0.3 V for 20 min to further remove O₂.

Figure R6. Experimental setup of *in-situ* electrochemical Raman spectroscopy.

3. In experiments using other electrolytes, were OH radicals also found?

Responses 3:

- We thank the reviewer for this interesting question. To answer this question, we supplemented further experiments. KCl, KI, and KClO₄ have been checked and we did not observe any signals of DMPO-OH in these electrolytes, as shown in Figure R7.

Figure R7. EPR spectra of electrolytes 0.5M KCl, 0.5M KI, and 0.1M KClO₄ (lower solubility for KClO₄).

4-Could the authors explain why the formation of OH radicals decreased when concentration of HCO₃ increased beyond 0.1 M?

Responses 4:

- We thank the reviewer for this question. When increasing KHCO₃ concentration, both the concentration of K⁺ and HCO₃⁻ have been increased. When the HCO₃⁻ concentration increases beyond 0.1 M, the high K⁺ concentration is likely to suppress the generation of DMPO-OH because, when we kept the K⁺ concentration at 0.5 M, then we found the order of DMPO-OH intensity: 0.2 M < 0.4 M ≤ 0.5 M (Figure R8).
- We supplemented the discussion in the main text on page 10.

“It is worth noting that further increasing the HCO₃⁻ concentration enhances the K⁺ concentration as well. Thus, at the same K⁺ concentration (0.5 M) with K⁺ compensation by K₂SO₄, the relationship between the HCO₃⁻ concentration and the intensity of DMPO-OH is more pronounced following the order 0.2 M < 0.4 M ≤ 0.5 M (Supplementary Fig. 10), and the intensity of DMPO-OH tends to saturate. The optimal HCO₃⁻ concentration without K⁺ compensation for the OH[•] radical formation is around 0.1 M (Fig. 5b).”

Figure R8 (Supplementary Fig. 10). EPR spectra of the KHCO₃/K₂SO₄ mixed solutions. The solutions at different HCO₃⁻/SO₄²⁻ mole ratios contain 100 mM DMPO under the same K⁺ concentrations of 0.5 M.

Minor:

5-In the Introduction, the authors stated in the 4th-5th line that partially-charged Cu δ⁺ species is likely to play a crucial role. I find that this is a misleading statement. As mentioned in my earlier review, this is a statement that has not been fully verified.

Responses 5:

We thank the reviewer for careful reading and detailed suggestions. We agree with the reviewer that it has not reached a consensus on whether Cuδ⁺ species is the key to CO-CO coupling. Therefore, we revised the discussion in the Introduction. Please find the revision on page 2 in the main text as follows:

“This is likely due to the partially charged $Cu\delta^+$ species that plays a crucial role.” has been changed into *“The precise mechanism remains unknown and different views have been proposed.”*

REVIEWERS' COMMENTS

Reviewer #1 (Remarks to the Author):

The author has addressed my questions. This research is qualified for publication.

Reviewer #3 (Remarks to the Author):

I thank the authors for revising their manuscript and performing extra experiments. I find the explanations to be reasonable. I am able to recommend this manuscript to Nature Comm.